JCB Journal of Cell Biology

# Developmental pruning of sensory neurites by mechanical tearing in *Drosophila*

Rafael Krämer[1]*, Neele Wolterhoff[1]*, Milos Galic[2], and Sebastian Rumpf[1]

**Mechanical forces actively shape cells during development, but little is known about their roles during neuronal morphogenesis. Developmental neurite pruning, a critical circuit specification mechanism, often involves neurite abscission at predetermined sites by unknown mechanisms. Pruning of *Drosophila* sensory neuron dendrites during metamorphosis is triggered by the hormone ecdysone, which induces local disassembly of the dendritic cytoskeleton. Subsequently, dendrites are severed at positions close to the soma by an unknown mechanism. We found that ecdysone signaling causes the dendrites to become mechanically fragile. Severing occurs during periods of increased pupal morphogenetic tissue movements, which exert mechanical forces on the destabilized dendrites. Tissue movements and dendrite severing peak during pupal ecdysis, a period of strong abdominal contractions, and abolishing ecdysis causes non-cell autonomous dendrite pruning defects. Thus, our data establish mechanical tearing as a novel mechanism during neurite pruning.**

## Introduction

Neurite pruning, the regulated loss of axons or dendrites during development without loss of the parent neuron, serves to specify neuronal connections and to remove developmental intermediates during neuronal development (Riccomagno and Kolodkin, 2015; Yu and Schuldiner, 2014) and often involves apparent neurite breaking or fragmentation. The peripheral sensory class IV dendritic arborization (c4da) neurons of *Drosophila* larvae specifically prune their dendrites during metamorphosis through proximal severing close to the cell body (Kuo et al., 2005; Williams and Truman, 2005). Dendrite pruning is induced by an ecdysone pulse at the onset of metamorphosis that induces the expression of pruning factors such as the transcription factor Sox14 and the actin severing enzyme Mical in a cell autonomous manner (Kirilly et al., 2009). These pruning factors lead to disassembly of actin (Kirilly et al., 2009; Wolterhoff et al., 2020) and microtubules (Herzmann et al., 2017; Lee et al., 2009; Williams and Truman, 2005) as well as locally increased endocytosis (Kanamori et al., 2015) in proximal dendrites. Microtubule dynamics in dendrites are increased during the early pupal stage by fine-tuned activation of the kinase Par-1 (Bu et al., 2022; Herzmann et al., 2017). The "plus end-in" orientation of dendritic microtubules is a spatial cue that localizes these degenerative processes to proximal parts of the dendrites (Herzmann et al., 2018; Rumpf et al., 2019; Wang et al., 2019). Once the dendrites are weakened in this way, they are severed from the cell body during an ill-defined period after 5 h after

puparium formation (APF), fragmented in a caspase-dependent manner (Williams et al., 2006), and phagocytosed by surrounding epidermal cells (Han et al., 2014).

While the cytoskeletal and trafficking mechanisms leading up to dendrite severing have been well characterized, the actual severing mechanism is unclear, apart from a spatial correlation with the local tissue architecture. Specifically, c4da neuron cell bodies and proximal primary dendrites are ensheathed by glia (Han et al., 2011), whereas the distal dendrites are sandwiched between the extracellular matrix and epidermal cells (Han et al., 2012; Kim et al., 2012). At some sites, contact between dendrites and epidermal cells is very close, such that dendrites seem to "sink into" the epidermal layer. It is estimated that 5–10% of the c4da dendritic arbor is enclosed in the epidermis in this way (Han et al., 2012; Kim et al., 2012). The eventual dendrite severing sites often coincide with the boundaries of glial ensheathment (Han et al., 2011).

The cytoskeletal alterations during the pruning process are predicted to alter neuronal mechanical properties (Spedden et al., 2012). Mechanophysical processes are well documented as important players during morphogenesis in a wide array of tissues (Dreher et al., 2016; Pasakarnis et al., 2016), and their role during neuronal morphogenesis is also becoming apparent (Franze, 2020). However, nothing is known about the actual role of mechanobiological processes during neuronal pruning. Here, we investigated mechanical aspects during c4da neuron

[1]Institute for Neurobiology, University of Münster, Münster, Germany; [2]Institute of Medical Physics and Biophysics, University of Münster, Münster, Germany.

*R. Krämer and N. Wolterhoff contributed equally to this paper. Correspondence to Sebastian Rumpf: sebastian.rumpf@uni-muenster.de

N. Wolterhoff's current affiliation is Division of Neurobiology, Institute for Biology, Freie Universität Berlin, Berlin, Germany.

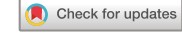

dendrite pruning. We showed that c4da neuron dendrites become mechanically fragile during the early pupal stage. Using live imaging, we found that c4da neuron dendrite severing occurs during periods of increased morphogenetic tissue movements that cause pulling forces on the weakened dendrites. Dendrite severing is especially prominent during pupal ecdysis, a morphogenetic process involving strong body contractions, and ablating ecdysis causes non-cell autonomous pruning defects. Thus, our data suggest that mechanical tearing contributes to developmental dendrite severing.

## Results

### C4da neuron dendrites become mechanically fragile during pruning

Larval peripheral neuron dendrites are exposed to constant stretch and compression during larval crawling (He et al., 2019; Vaadia et al., 2019) and must therefore be highly mechanically stable. Yet, these dendrites are selectively pruned through an apparent breaking or severing mechanism at the onset of the pupal stage. During this process, the cytoskeleton is disassembled in proximal dendrites prior to severing (Bu et al., 2021; Herzmann et al., 2018; Kirilly et al., 2009; Rumpf et al., 2019; Wang et al., 2019; Wolterhoff et al., 2020). Cytoskeletal disassembly occurs in response to ecdysone signaling, but a direct link had previously only been known for actin, as expression of the actin severing enzyme Mical is induced by ecdysone (Kirilly et al., 2009). The kinase regulating microtubule dynamics during pruning, Par-1, is known to be activated by active site phosphorylation, and the Par-1 upstream kinase LKB1 is also required for dendrite pruning (Marzano et al., 2021). In support of a direct effect of ecdysone also on microtubule regulation, we found that ecdysone treatment increased Par-1 levels and active site phosphorylation in biochemical experiments in S2 cells (Fig. S1).

This concerted cytoskeleton disassembly is predicted to drastically alter the viscoelastic and mechanical properties of c4da neuron dendrites. We therefore wanted to assess the stability of c4da neuron dendrites during the early pupal stage. As the presence of the hardening pupal case precludes any direct quantitative methods during the early pupal stage, we applied mechanical stress by brief sonication in an ice-cooled water bath. At the white pupal stage (0 h APF), exposure to 30-s bouts of sonication in a water bath did not cause apparent damage to dendrites of neurons expressing a control dsRNA against the odorant coreceptor Orco (which is not expressed in c4da neurons and not required for PNS development; Fig. 1, A, A', and F). However, at 5 h APF, when the dendrites had developed local thinnings due to the influence of ecdysone on the cytoskeleton (Fig. 1 E), sonication caused new breaks in dendrites in almost half of the analyzed neurons. These new breaks were mostly localized in the thinned primary or secondary dendrites (Fig. 1, C, C', F, and G). In order to test whether this enhanced sensitivity to mechanical stress was a consequence of ecdysone signaling, we next knocked down Sox14, the key mediator of ecdysone during pruning (Kirilly et al., 2009), in c4da neurons. Sox14 dsRNA did not affect dendrite stability at the onset of the

pupal stage (Fig. 1, B, B', and F). However, it greatly diminished the number of sonication-induced breaks in proximal dendrites at 5 h APF (Fig. 1, D and F–G), indicating that they were more stable than dendrites of control neurons. We concluded that ecdysone signaling destabilizes c4da neuron dendrites mechanically.

### Evidence for mechanical tension during dendrite severing

To better understand the mechanism of initial severing of proximal dendrite regions during c4da neuron dendrite pruning, we next imaged c4da neurons live during 1-h windows between 5 and 12 h APF in 1-min intervals using an inverted confocal microscope and looked for severing events in primary or secondary dendrites close to the c4da neuron soma. Surprisingly, recorded severing events often showed evidence of mechanical stress. These included snap-back movements of severed dendrite ends (Fig. S2, A and B; and Video 1), indicative of pulling forces, and tissue movements immediately before severing (Fig. S2 C and Video 2). To explore a potential role for extrinsic forces during severing more systematically, we imaged c4da neurons in 5-min intervals over several hours starting around pupariation (Fig. 2, A–A‴ and Video 3). These analyses revealed frequent, relatively slow movements of c4da neuron cell bodies and the surrounding epidermal tissue, especially after 5 h APF, as shown by tracing c4da neuron cell bodies in the time lapse movies (Fig. 2, A and B; and Video 3). The occurrence of dendrite severing events clearly increased after the onset of these tissue movements (Fig. 2, A‴ and B and Video 3).

C4da neuron dendrites are in close contact with epidermal cells (Han et al., 2012; Kim et al., 2012) while their cell bodies are wrapped by glia, and the boundary between epidermis and glia in the proximal dendrites often marks severing sites (Han et al., 2011). In order to address whether the observed tissue movements could affect cell bodies and dendrites differentially, we decided to measure their relative distances in our time lapse analyses. To this end, we labeled c4da neurons with *ppk-eGFP* and surrounding epithelial cells with *ECad::GFP*. We then imaged c4da neurons and epidermis in 5-min intervals between 0 and 8 h APF and traced the positions of (1) neuronal cell bodies, (2) clearly recognizable dendritic branchpoints, and (3) tricellular junctions of adjacent epidermal cells (Fig. 2, C and C' and Video 4). The whole tissue underwent movements in these experiments, but the distance between the dendritic and epidermal landmarks stayed largely constant (Fig. 2, C and D), indicating that they moved synchronously. In contrast, the distance between the c4da neuron cell body and the epidermal landmark started to change early, and even more strongly after ~5 h APF (Fig. 2, C and D), indicative of asynchronous movements (e.g., due to slight differences in speed and/or movement direction). These tendencies—synchronous movement between dendrite and epidermis and asynchronous movement between cell body and epidermis—were seen consistently in such time-lapse analyses, especially after 5 h APF (Fig. 2 E).

C4da neuron dendrites extend both along the anterior-posterior axis and the dorso-ventral axis. During the time of the experiments (i.e., until the time of severing), cell bodies and dendrite landmarks could diverge (or move closer to each other)

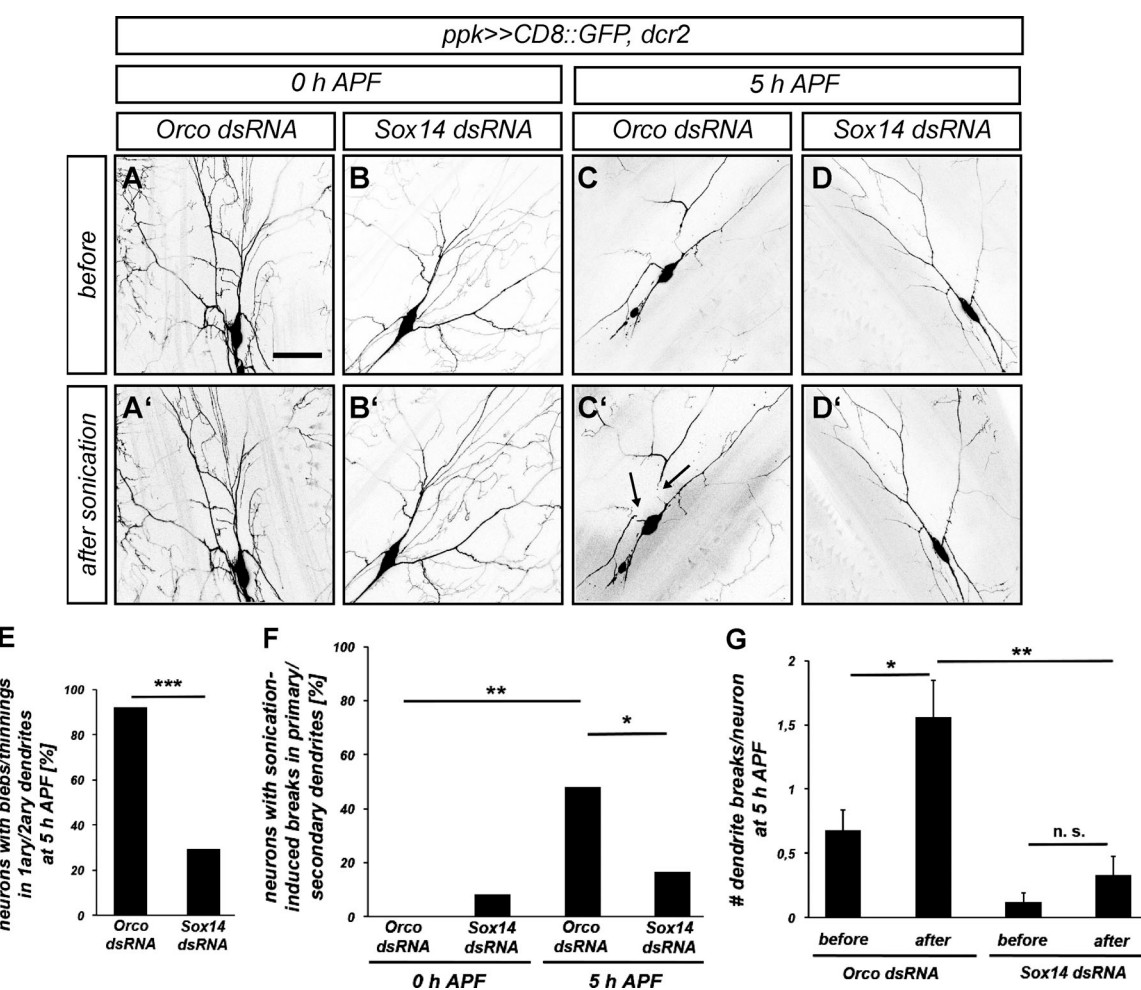

Figure 1. **Mechanical sensitivity of early pupal dendrites. (A–D)** Mechanical stability of dendrites of white pupal (0 h APF [A–B']) and pupal (5 h APF [C–D']) dorsal c4da neurons as assessed by sensitivity to brief sonication in a water bath. A–D show dorsal c4da neurons before sonication, A'–D' show the same neurons after 30 s sonication. **(A and A')** White pupal control c4da neuron expressing Orco dsRNA. **(B and B')** White pupal c4da neuron expressing Sox14 dsRNA. **(C and C')** Control c4da neuron expressing Orco dsRNA at 5 h APF. **(D and D')** C4da neuron expressing Sox14 dsRNA at 5 h APF. **(E)** Quantification of neurons with proximal dendrite regions containing thinnings and blebs in C and D. N = 24–25, ***P < 0.0005, two-tailed Fisher's exact test. **(F)** Quantification of sonication-induced breaks in proximal primary and secondary dendrites. N = 24–25, *P < 0.05, **P < 0.005, two-tailed Fisher's exact test. **(G)** Total number of breaks in proximal primary and secondary dendrites at 5 h APF before and after sonication. Values are mean ± SEM, N = 24–25, *P < 0.05, **P < 0.005, Mann–Whitney U test. Scale bars in A and C are 50 μm.

along both the anterior-posterior and the dorso-ventral axis (Fig. S3). As a consequence of such differential movement, the connecting proximal dendrites could experience mechanical stress such as stretch and pulling forces. Indeed, the actual length of the analyzed dendrites between the soma and the dendritic landmark in our live imaging analysis often seemed to increase between the onset of the pupal stage and the time of severing (Fig. 2, D and F). We therefore specifically identified dendrites that became severed in our time-lapse movies and measured the lengths of their proximal regions between the soma and the second branchpoint along the primary dendrite. Almost all of these dendrites became longer between the onset of the experiment and the time of severing with an average length increase of more than 10%, indicating that they were stretched over time (Fig. 2 G).

Pulling forces on parental neurite branches at neurite branchpoints are known to decrease the angle between the

daughter branches (Bray, 1979). Hence, stretch relaxation (e.g., through cutting of the parental branch) often results in increased branching angles (Bray, 1979). Indeed, the branching angles at branchpoints distal to severing sites often increased between the onset of metamorphosis and after dendrite severing in our live imaging analyses (Fig. 2, H and I), indicative of such stretch relaxation. Taken together, our data suggest that during the early pupal stage, mechanically destabilized c4da neuron dendrites are exposed to stretch forces caused by the differential effects of tissue movements on the cell bodies and distal dendrites.

## Dendrite severing and pupal body movements increase during pupal ecdysis

In parallel to the above experiments, we established a systematic timecourse of dendrite severing by counting the number of primary dendrites attached to the soma of c4da neurons (Fig. 3).

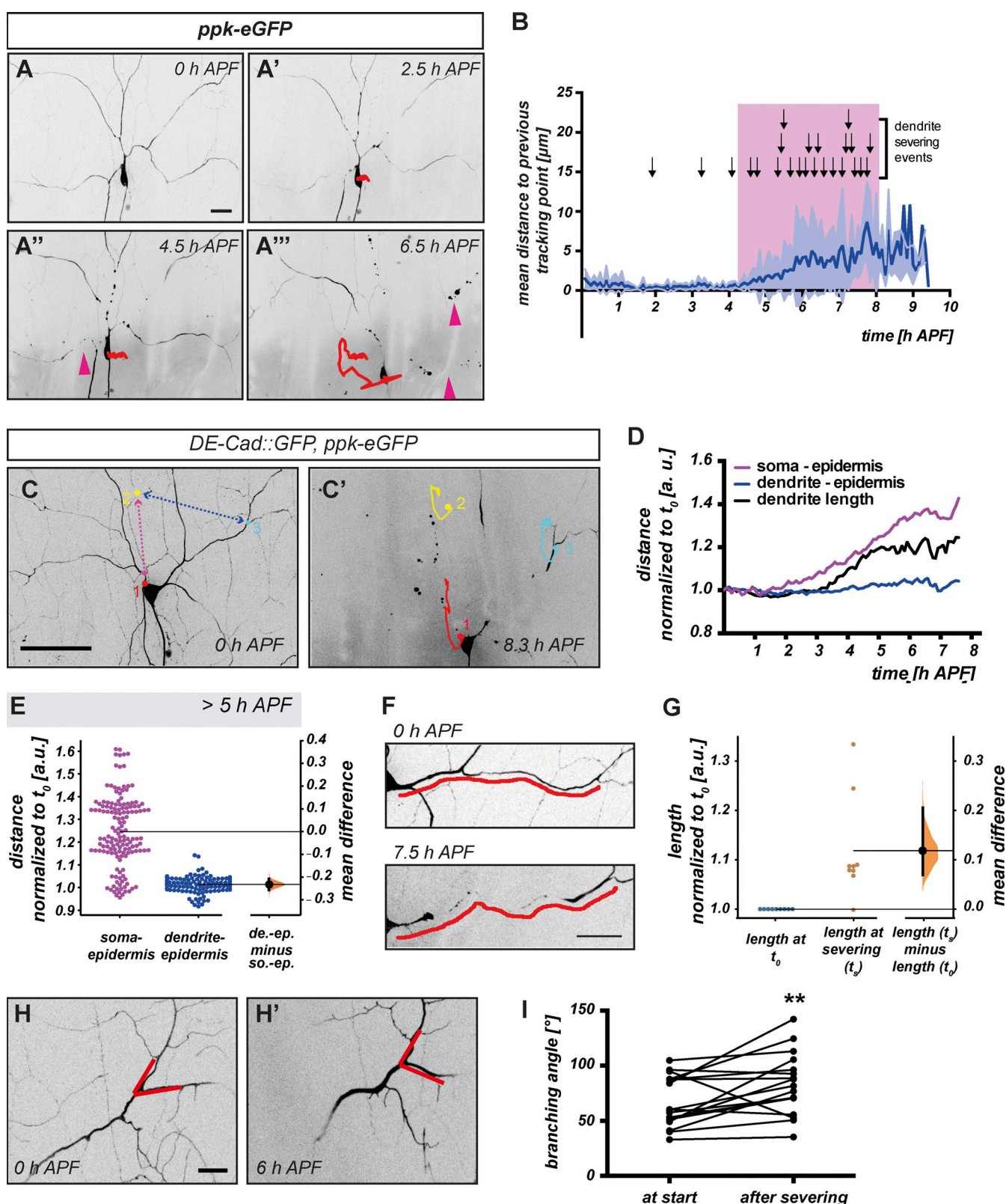

**Figure 2.** **Evidence that pupal tissue movements exert force on c4da neuron dendrites. (A and B)** Time-lapse analysis of c4da neuron dendrite severing and correlation with tissue movements during the early pupal stage. A–A‴ are stills from a representative time-lapse movie (1 frame every 5 min between 0–6.5 h APF) of a dorsal c4da neuron labeled by *ppk-eGFP* (see also Video 3). The colored trace shows cell body position over the analysis, with color-coded timepoints. Red arrowheads mark severed dendrites. **(B)** Graph depicting average movement magnitude over time from 13 time-lapse movies. Error envelope represents standard deviation, arrows over graph indicate dendrite severing events. **(C–I)** C4da neuron cell bodies and distal dendrite branchpoints (labeled by *ppk-eGFP*) as well as epidermal cell tricellular junctions (labeled by *DE-Cad::GFP*) were traced in time-lapse movies (e.g., Video 4) starting at 0 h APF. **(C)** C4da

neuron at 0 h APF (start of the analysis) with cell body (red), dendrite (blue) and epidermal landmarks (yellow) indicated. **(C')** The same neuron at the end of the time-lapse experiment, with the traces of the respective landmarks indicated. **(D)** Graph showing distances between the dendrite branchpoint or the cell body and the epidermis landmark in C–C' to assess synchronicity of movement over time. The black curve shows dendrite length (between soma and dendrite branchpoint landmark) over time. All values were normalized to the initial timepoint. **(E)** Distribution of cell body/epidermis and dendrite/epidermis distances after 5 h APF normalized to the distance at $t_0$ ($N$ = 10 time-lapse movies). **(F)** The dendrite segment between cell body and dendrite landmark from time-lapse movie in C–C'. Upper panel shows the dendrite at 0 h APF, the lower panel shortly before severing (7.5 h APF). The segment is marked with a red line for length comparison. **(G)** In these time-lapse movies, the length of primary dendrite proximal segments (between soma and second branchpoint) was measured at the onset of the pupal stage ($t_0$) and shortly before severing. $N$ = 9, values are normalized to $t_0$. **(H and H')** Image showing a dendrite branch point distal to an eventual severing site at the onset of metamorphosis (H) and after severing (H'). The branching angle is indicated by red lines. **(I)** Graph depicting branching angles at branchpoints distal to severing sites at the start of metamorphosis (0–90 min APF) and after severing (250–515 min APF). \*\*$P$ < 0.005, Mann–Whitney U test ($N$ = 14). Scale bars in A, C, and H are 20, 50, and 10 μm, respectively.

During the wandering third instar larval stage, dorsal c4da neurons (ddaC) in the abdominal segments A2–A5 have an average of three primary dendrites (Fig. 3, A and G). Over the first 10 h after puparium formation (h APF), dendrite severing occurred in a slow and stochastic manner, and c4da neurons still had approximately two attached primary dendrites at 10 h APF (Fig. 3, B–D and G). Most primary dendrites were severed between 10 and 12 h APF, such that at 12 h APF, an average of <0.5 dendrites were still attached per neuron (Fig. 3, E–G). This distinct pattern of peak dendrite severing between 10 and 12 h APF was also seen in the ventrolateral c4da neuron v'ada (Fig. 3, H–N). Class I da neurons (dorsal neurons ddaD and ddaE) are a second type of peripheral sensory neurons which also prune their dendrites during metamorphosis. While severing events in this type of neuron seemed more randomly distributed along the entire dendrite compared to c4da neurons (Fig. 3 S), most severing occurred again between 10 and 12 h APF (Fig. 3, O–U).

The 10–12 h APF window coincides with pupal ecdysis, a morphogenetic process that leads to eversion of the head, legs, and wings and involves strong abdominal contractions. To assess a potential link between ecdysis, morphogenetic tissue movements and dendrite pruning, we extended our time-lapse analysis beyond 10 h APF (Fig. 4). This showed that the movement of c4da neuron cell bodies and the surrounding tissue strongly increased after ~9 h APF (Fig. 4, A and D; and Video 5). Ecdysis depends on the ETS (E26 transformation-specific) domain transcription factor E74 (Fletcher et al., 1995) and on a peptidergic neuronal cascade including neurons expressing *Crustacean Cardioactive Peptide* (CCAP; Park et al., 2003). We confirmed that loss of E74 and CCAP neuron ablation (through expression of the cell death gene *head involution defective* (*hid*) under *CCAP-GAL4*) causes strong head eversion defects (Fig. S4). We then traced c4da neuron cell bodies in *E74DL-1* mutant animals and in animals with ablated CCAP neurons during the prepupal stage. In both *E74DL-1* mutant animals and upon CCAP neuron ablation, the earlier cell body movements were of similar amplitude as in controls, but the increase between 10 and 12 h APF was completely abolished (Fig. 4, B–D and Videos 6 and 7).

As our stretch analysis had focused on a time period before ecdysis, we wished to independently assess whether these ecdysis-induced movements could also induce mechanical stress in the epidermis surrounding the dendrites. Because the ecdysis movements were faster than the earlier tissue movements, we co-labeled epidermal cell mitochondria with *UAS-mito::GFP* and c4da neurons with *ppk-CD4::tdtomato* and imaged single confocal

planes containing c4da neuron cell bodies and dendrites in 1-s intervals. We then used spatiotemporal image correlation (Ashdown et al., 2015) of patterns of epidermal mitochondria to generate velocity histograms of the movement of the epidermis cells. At an earlier developmental timepoint around 7 h APF, the movement amplitude at this time scale was small (<20 μm/min, compare color code for velocities; Fig. 4, E and E'), but the movement became much more vigorous at 10 h APF and reached velocities of over 300 μm/min (Fig. 4, F and F', and Video 8). Importantly, local differences in movement speed and directionality in closely adjacent areas could be observed at 10 h APF (i.e., different colors and directions of arrows in Fig. 4, F and F'), indicating that ecdysis-induced movements also generate local stretch and shear in the vicinity of c4da neuron cell bodies and dendrites.

## Ecdysis inhibition causes non-cell autonomous dendrite pruning defects

To establish a causal relationship between dendrite severing and ecdysis movements, we next asked whether ecdysis manipulations cause pruning defects. C4da neurons in heterozygous *E74DL-1*/+ control animals had completely pruned their dendrites at 16 h APF (Fig. 5, A and C). In contrast, a significant fraction of c4da neurons in *E74DL-1/v4* mutants still had dendrites attached to the cell body at this time (Fig. 5, B and C). In these animals, proximal regions of c4da neuron dendrites clearly had thinnings and varicosities, indicating that cell autonomous cytoskeleton disassembly was unaffected. In addition, severed but unfragmented long dendrites were often still present in the mutant at this timepoint, possibly indicative of defects in post-severing fragmentation (Fig. 5, A, B, and D).

If the observed pruning defects in the *E74* mutant were due to defects in ecdysis and head eversion, E74 would be expected to act in a non-cell autonomous manner. In support of this notion, expression of the cell autonomous ecdysone targets Sox14 and Mical in the c4da neurons was not altered in *E74DL-1/v4* mutant animals at 2 h APF (Fig. 5, E–F' and G–H'). We next knocked down E74 specifically in c4da neurons under the control of *ppk-GAL4*. Consistently, this did not cause detectable pruning defects at 14 h APF (Fig. 5, I, L, and M). In order to be able to knock down E74 in the whole animal with the exception of c4da neurons, we combined the ubiquitous driver *act5C-GAL4* with a *ppk-GAL80* transgene (Fig. S4). Expression of a control dsRNA construct against the odorant coreceptor Orco under this driver did not cause ecdysis defects (Fig. S4), and c4da neurons in these

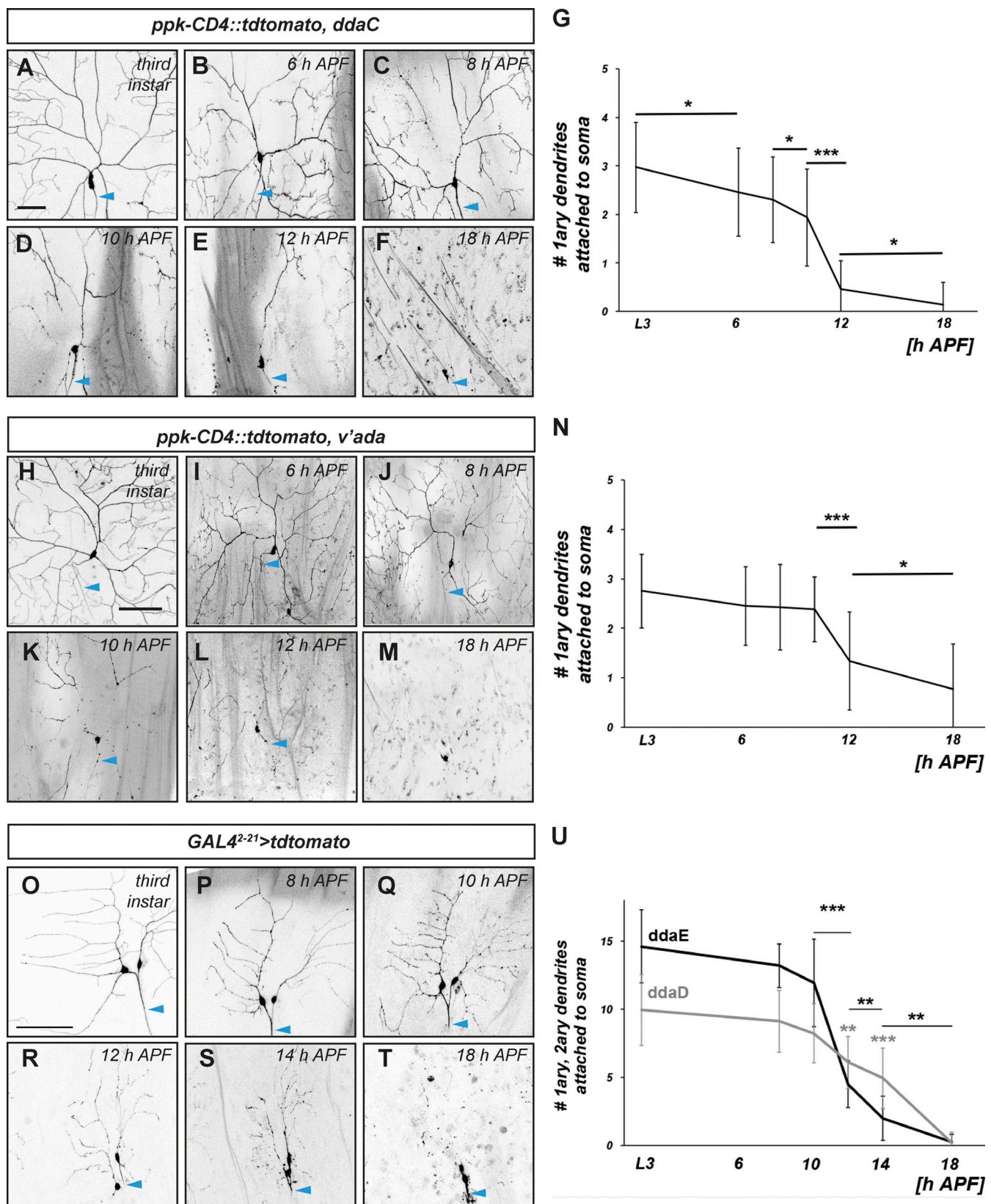

Figure 3. **Timecourse analysis of PNS neuron dendrite severing. (A–F)** Representative images of dorsal ddaC c4da neurons (visualized by ppk-CD4::tdtomato) at the indicated developmental timepoints. **(G)** Graph depicting the average number of primary dendrites attached to the c4da neuron cell body at the indicated timepoints. Values are mean ± SD, *P < 0.05, ***P < 0.0005, Mann–Whitney U test, N = 22–63 neurons per timepoint. **(H–M)** Representative images of v'ada c4da neurons at the indicated developmental timepoints. **(N)** Graph depicting the average number of primary dendrites attached to the v'ada

neuron cell body at the indicated timepoints. Values are mean ± SD, *P < 0.05, ***P < 0.0005, Mann–Whitney U test. N = 22–30 neurons per timepoint. **(O–T)** Timecourse of dendrite severing during dendrite pruning of dorsal c1da neurons ddaD (right) and ddaE (left; visualized by expression of UAS-tdtomato under the control of *GAL4$^{2-21}$*). Panels show representative images of c1da neurons at the indicated developmental timepoints. **(U)** Graph depicting the average number of primary and secondary dendrites attached to the cell body of ddaE and ddaD at the indicated timepoints. Values are mean ± SD, **P < 0.005, ***P < 0.0005, Wilcoxon test. *N* = 9–14 neurons per timepoint. Blue arrowheads in panels indicate axons. Scale bars are 50 μm in A and 100 μm in H and O.

animals had normally pruned their dendrites (Fig. 5, J, L, and M). Ubiquitous expression of E74 dsRNA caused ecdysis defects (Fig. S4), and as in *E74$^{DL-1/v4}$* mutant animals, c4da neurons in these animals showed pruning defects with dendrites that were still attached to the cell body (Fig. 5, K–M). Again, proximal regions

of such dendrites often showed thinnings and varicosities (Fig. 5 K).

As with E74 mutants, CCAP neuron ablation did not affect ecdysone target gene expression in c4da neurons (Fig. 6, A–D'). In animals with ablated CCAP neurons, dendrites of dorsal ddaC

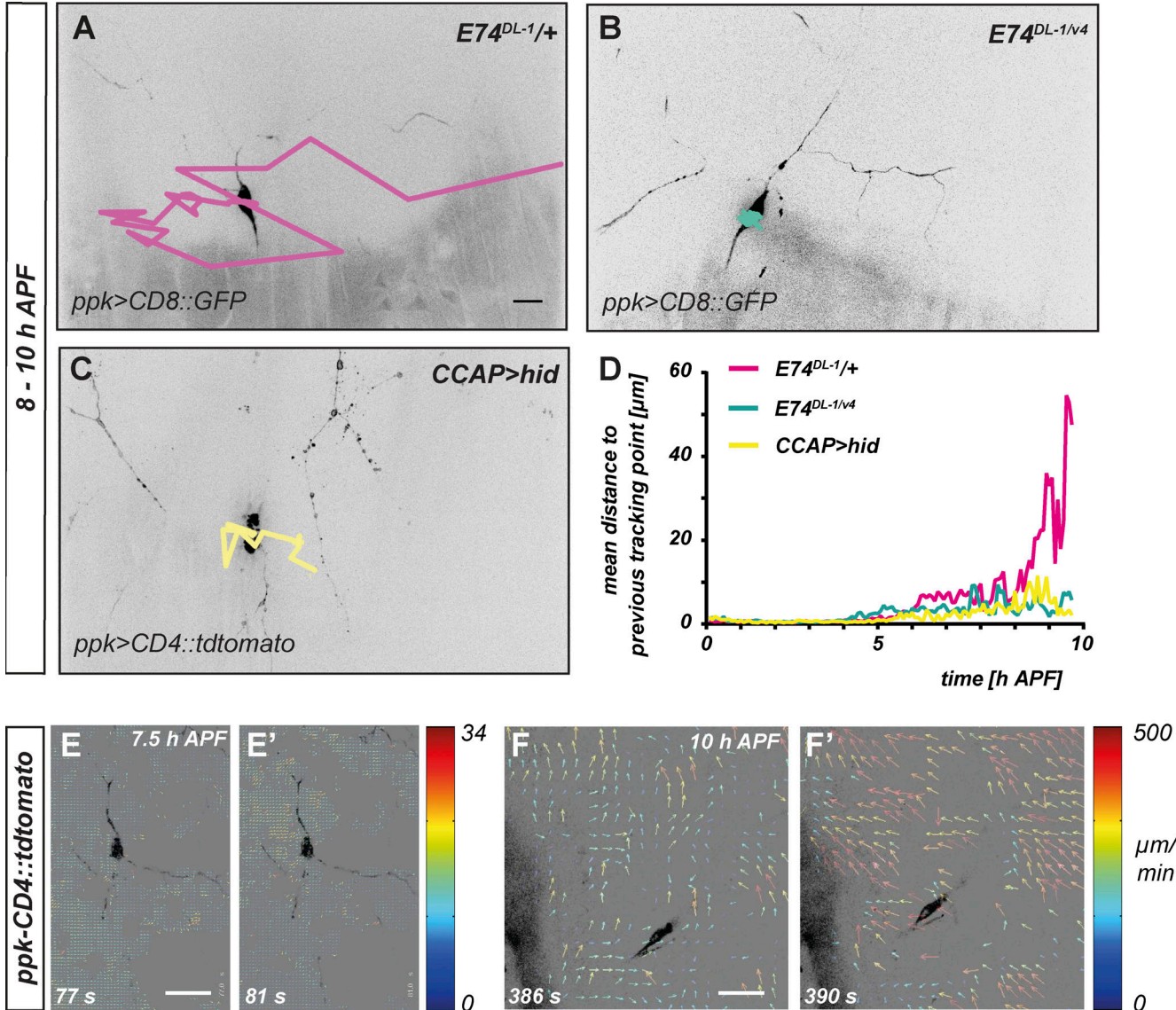

Figure 4. **Morphogenetic movement of c4da neuron cell bodies after 8 h APF is linked to pupal ecdysis. (A–C)** c4da neuron cell body tracings in time lapse movies (5-min intervals) of animals of the indicated genotypes between 8 and 10 h APF. **(A)** Heterozygous *E74$^{DL-1}$/+* control. **(B)** *E74$^{DL-1/v4}$* mutant. **(C)** Animal with ablated CCAP neurons (*hid* expression under *CCAP-GAL4*). **(D)** Graph depicting amplitude of c4da neuron cell body movements of the genotypes in B–D during the early pupal stage. Traces show average displacement from previous position in μm (*N* = 8–10 neurons each) over time (h APF). **(E–F')** Spatiotemporal image correlation analysis of tissue movement. Arrows depict velocity of local tissue movement as determined by tracing epidermal mitochondria in the vicinity of the c4da neuron (see color code for movement velocity) in 1-s intervals in single confocal planes. E and E' show representative frames of a c4da neuron at 7.5 h APF; F and F' show representative frames of a c4da neuron at 10 h APF.

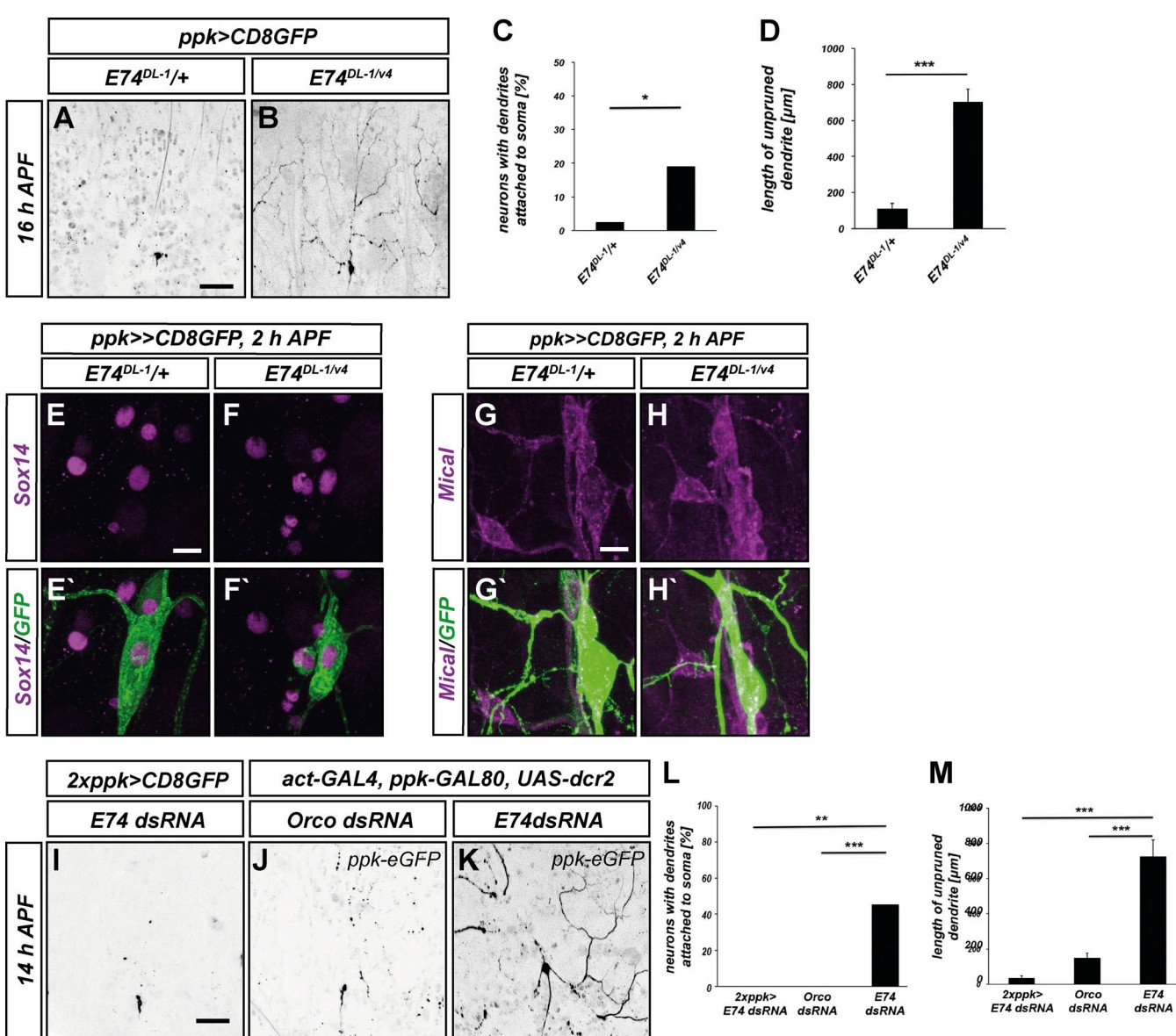

Figure 5. **Loss of E74 causes non-cell autonomous c4da neuron pruning defects. (A–D)** Dendrite pruning defects in a E74 mutant. **(A)** C4da neuron labeled by *CD8::GFP* expressed under *ppk-GAL4* in heterozygous *E74^{DL-1}/+* animal at 16 h APF. **(B)** C4da neuron in *E74^{DL-1/v4}* mutant animal at 16 h APF. **(C)** Penetrance of dendrite severing defects (fraction of neurons with dendrites attached to soma) in samples A and B. N = 68, 39, *P < 0.05, two-tailed Fisher's exact's test. **(D)** Total length of unpruned dendrites in samples A and B. Values are mean ± SEM, ***P < 0.0005, Wilcoxon's test. **(E–H')** C4da neurons were labeled by CD8::GFP expression under the control of *ppk-GAL4*, and immunofluorescence of the pruning factors Sox14 and Mical was performed at 2 h APF. **(E–F')** Sox14 expression in *E74^{DL-1}/+* heterozygous controls (E and E') or in *E74^{DL-1/v4}* mutants (F and F'). **(G–H')** Mical expression in *E74^{DL-1}/+* heterozygous controls (G and G') or in *E74^{DL-1/v4}* mutants (H and H'). **(I–M)** Non-cell autonomous requirement for E74 during dendrite pruning. **(I)** E74 dsRNA was expressed in c4da neurons under *ppk-GAL4*, and dendrite pruning was assessed at 14 h APF. **(J and K)** *Orco* control dsRNA (J) or *E74* dsRNA (K) were expressed ubiquitously under *act5C-GAL4* in the presence of *ppk-GAL80* to exclude c4da neurons, and dendrite pruning was assessed at 14 h APF. **(L)** Penetrance of dendrite severing defects in I–K. N = 32, 25, 21, **P < 0.005, ***P < 0.0005, two-tailed Fisher's exact test. **(M)** Total length of unpruned dendrites in I–K. Values are mean ± SEM, ***P < 0.0005, Wilcoxon's test.

c4da neurons were mostly severed at 18 h APF, but detached, unfragmented dendrites persisted, possibly indicative of delayed severing as well as fragmentation defects (Fig. 6, E and F, see also Fig. 7, B, D, and E for more quantification). In contrast to these relatively mild defects, CCAP neuron ablation caused strong severing defects in the ventrolateral (v'ada) and ventral (vdaB) c4da neurons at 18 h APF (Fig. 6, E–H). Microtubule stainings still showed gaps in v'ada neuron proximal dendrites in CCAP-

ablated at 5 h APF, again suggesting that CCAP neuron ablation did not affect cell autonomous processes (Fig. S5). Interestingly, it was shown recently that CCAP neuron inhibition only blocks a specific part of the ecdysis motor program (Elliott et al., 2021), such that locally, some movements may be more affected by CCAP ablation than others. In support of a general role of ecdysis during dendrite severing, CCAP neuron ablation also led to the retention of unpruned, attached dendrites in the c1da neuron

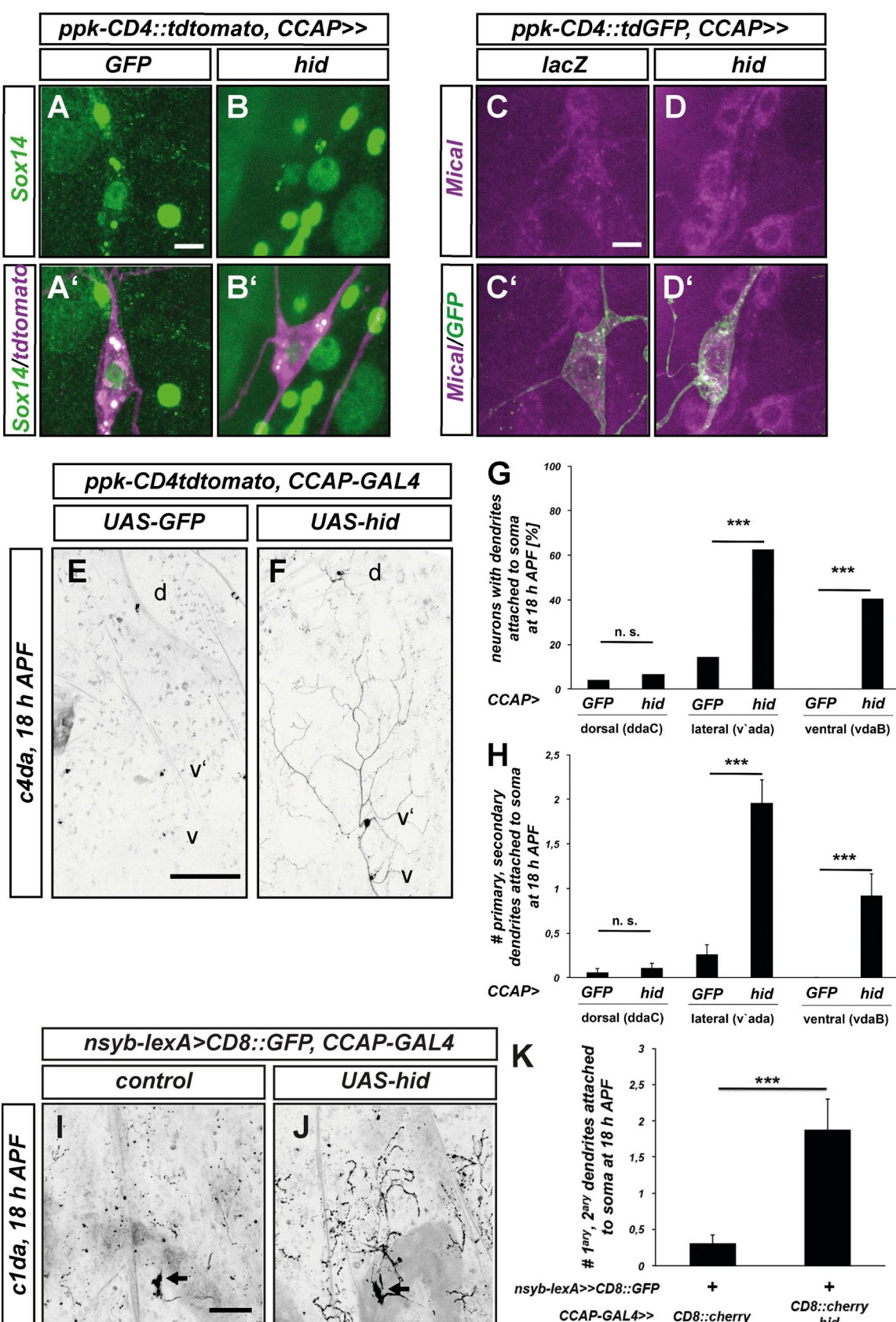

Figure 6. **CCAP neuron ablation causes PNS neuron dendrite pruning defects. (A–D')** Effect of CCAP neuron ablation on ecdysone target gene expression in c4da neurons. C4da neurons were labeled by *ppk-CD4::tdtomato* or *ppk-CD4::tdGFP* in control animals (*CCAP > GFP*, *CCAP > lacZ*) or in animals with ablated CCAP neurons (*CCAP > hid*), and immunofluorescence of Sox14 and Mical was performed at 2 h APF. **(A–B')** Sox14 expression in *CCAP > GFP* controls (A and A')

and in *CCAP*-ablated animals (B and B'). **(C–D')** Mical expression in *CCAP > lacZ* controls (C and C') and in *CCAP*-ablated animals (D and D'). **(E–H)** Dendrite severing defects in lateroventral and ventral c4da neurons upon ablation of CCAP neurons. C4da neurons were labeled by *ppk-CD4::tdtomato*. Shown are hemisegments of pupae of the indicated genotypes at 18 h APF. Positions of the dorsal (d), lateroventral (v') and ventral (v) c4da neurons are indicated. **(E)** Control animal expressing GFP under the control of *CCAP-GAL4*. **(F)** Animal expressing *UAS-hid* under *CCAP-GAL4* to ablate CCAP neurons. **(G)** Penetrance of dendrite severing defects in E and F as shown by the percentage of c4da neurons with unpruned, attached dendrites at 18 h APF. n.s., not significant, *P < 0.05, ***P < 0.0005, two-tailed Fisher's exact test (N = 29–68). **(H)** Severity of dendrite severing defects in E and F as assessed by number of primary and secondary dendrites attached to the soma. Values are mean ± SEM, n.s., not significant, *P < 0.05, ***P < 0.0005, Wilcoxon's test. **(I–K)** C1da neuron pruning defects upon *CCAP* neuron ablation. Multidendritic sensory neurons were labeled by expression of CD8::GFP under the control of *nsyb-lexA*. C1da neurons were identified by cell body and dendritic morphology. Cell body positions of the c1da neuron ddaD are marked by arrows. **(I)** Dorsal sensory neurons in control animals carrying *CCAP-GAL4*. **(J)** Dorsal sensory neurons in animal with ablated *CCAP* neurons. **(K)** Quantification of dendrite severing defects as shown by the number of primary and secondary dendrites attached to the ddaD cell body at 18 h APF. Values are mean ± SEM, ***P < 0.0005, Wilcoxon test (N = 39, 25). Scale bars are 10 µm in A and C, 100 µm in E, and 50 µm in I.

ddaD at 18 h APF (Fig. 6, I–K). Taken together, we excluded that our ecdysis manipulations affect the cell autonomous ecdysone-dependent c4da neuron pruning program. While we cannot entirely exclude that they may cause defects in other unknown non-cell autonomous pruning pathways, the dendrite pruning defects observed upon ecdysis inhibition are most consistent with a role of tissue movements during dendrite severing.

### Interplay between extrinsic forces and local pruning pathways

The above data suggest a model where ecdysone-induced cytoskeleton disassembly weakens the proximal dendrites and predisposes them to severing by mechanical tearing. In this model, manipulations that increase dendritic cytoskeleton stability should interact synergistically with ecdysis manipulations during dendrite severing. To test this prediction, we performed CCAP neuron ablation in a background heterozygous for *swp2^MICAL*, a strong *mical* mutant (Beuchle et al., 2007; Terman et al., 2002). While *MICAL* heterozygosity did not cause dendrite pruning defects by itself (Fig. 7, A, D, and E), the combination with CCAP neuron ablation caused a significant increase in the number of neurons with dendrites attached to the cell body (Fig. 7, B–E).

After the severing step, c4da neuron dendrites are fragmented in a caspase-dependent manner (Williams et al., 2006) and then phagocytosed by epidermal cells that act as non-professional phagocytes (Han et al., 2014). To assess whether the severed, but not fragmented dendrites retained upon CCAP neuron ablation were in contact with epidermal autophagosomes, we used the pHluorin-CD4-tdtomato phagocytosis reporter *ppk-MApHS* (Han et al., 2014). In this reporter, the pHluorin moiety can only fluoresce when present at the cell surface, but not in acidic phagosomes. In control animals with intact CCAP neurons, pHluorin could only be detected in c4da neuron cell bodies, whereas c4da neuron-derived tdtomato, but not pHlourin fluorescence, could be detected in epidermal cells at 14 h APF, indicating localization in epidermal phagosomes (Fig. 7, F–F″ and H). In contrast, CCAP ablation led to retention of long stretches of unengulfed, pHluorin-positive dendrites (Fig. 7, G–G″ and H).

Phagocytosis by epidermal cells and the accompanying polymerization of epidermal actin around severed dendrites have been proposed to contribute to dendrite fragmentation (Han et al., 2014; Williams et al., 2006), but it was unclear whether this could only act on severed dendrites or also on attached ones.

A cursory investigation of F-actin structures in early pupal epidermis cells showed that F-actin-rich structures occurred around both severed dendrites and dendrites that were still attached to the soma (Fig. S5), opening up the possibility that local F-actin could also play a role during dendrite severing. Because of the similar phenotypes of the ecdysis and phagocytosis/fragmentation pathways, we next tested for genetic interactions. While combined heterozygous mutations in the phagocytosis factors Draper and WASp did not cause dendrite pruning defects in the dorsal ddaC neurons (Fig. 7, I, L, and M), and CCAP neuron ablation only caused accumulation of severed dendrite fragments (Fig. 7, J, L, and M), the combination of these manipulations led to a significant increase in the fraction of c4da neurons with dendrites attached to the cell body (Fig. 7, K–M). Thus, our data indicate that both cell autonomous and non-cell autonomous degenerative pathways cooperate closely at the severing step during dendrite pruning.

## Discussion

In this study, we assessed the role of external physical forces during c4da neuron dendrite severing and pruning. We first provided evidence that the cell autonomous ecdysone response in the neurons leads to mechanical weakening of proximal dendrites. We then showed that dendrite severing is associated with signs of mechanical stress induced by morphogenetic movements of the surrounding tissue. Finally, we identify contractile movements of the abdominal epidermis caused by pupal ecdysis as one cause of the morphogenetic movements. Our data indicate that loss of ecdysis leads to non-cell autonomous dendrite pruning defects that can be enhanced by manipulations of cell autonomous (cytoskeleton) and local non-cell autonomous (phagocytosis) pruning pathways. To test whether mechanical forces are sufficient to tear dendrites, the effects of optogenetic induction of ecdysis movements at various developmental stages on dendrites could be assessed.

Mechanical dendrite severing during pruning is likely aided by the local tissue architecture. Specifically, the boundaries of the glial ensheathment of proximal dendrites might act as rigid edges or counterbearings upon which dendrites break. Indeed, the use of tissue counterbearings has recently been described during insect gastrulation (Münster et al., 2019). Taken together, our work suggests that mechanical tearing is a novel mechanism during large scale pruning.

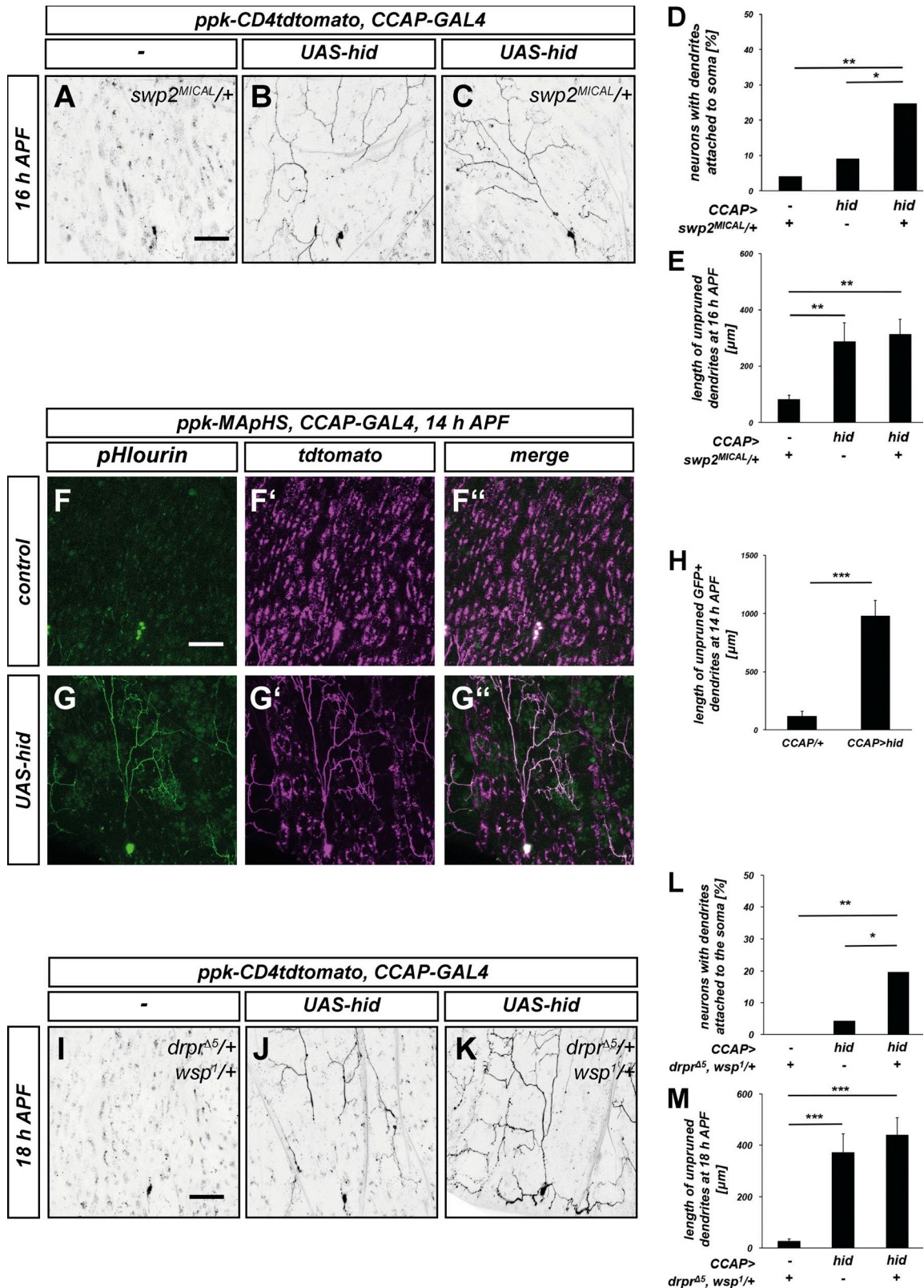

Figure 7. **Synergistic genetic interactions between ecdysis and local dendrite pruning pathways. (A–C)** C4da neurons were labeled with *ppk-CD4::tdtomato* and dendrite pruning phenotypes were assessed at 16 h APF. **(A)** Animal heterozygous for the *mical* mutant allele *swp2^MICAL*. **(B)** Animal expressing *UAS-hid* under *CCAP-GAL4* to ablate CCAP neurons. **(C)** *swp2^MICAL* heterozygote with ablated CCAP neurons. **(D)** Penetrance of neurons with dendrite severing defects in A–C. *P < 0.05, **P < 0.005, two-tailed Fisher's exact test, N = 48–69 neurons. **(E)** Length of unpruned dendrites in A–C. Values are mean ± SEM, **P < 0.005, Wilcoxon-Mann-Whitney test. **(F–H)** Ecdysis inhibition causes phagocytosis defects during dendrite pruning. C4da neurons were labeled by pHluorin-CD4-tdtomato expression under the *ppk* promotor (*ppk-MApHS*), and pHluorin/GFP and tdtomato fluorescence was assessed at 14 h APF. **(F–F")** C4da

neuron in control animal. **(G–G″)** C4da neuron in animal with ablated CCAP neurons. **(H)** Length of pHluorin-positive (severed and unsevered) dendrites in F and G. Values are mean ± SEM, ***$P < 0.0005$, Wilcoxon-Mann-Whitney test, $N = 27$ each. **(I–K)** C4da neurons were labeled with *ppk-CD4::tdtomato* and dendrite pruning phenotypes were assessed at 18 h APF. **(I)** Animal heterozygous for *drpr$^{\Delta 5}$, wsp$^1$* mutant alleles. **(J)** Animal expressing *UAS-hid* under *CCAP-GAL4* to ablate CCAP neurons. **(K)** *drpr$^{\Delta 5}$, wsp$^1$* heterozygote with ablated CCAP neurons. **(L)** Penetrance of dendrite severing defects in I–K. *$P < 0.05$, **$P <$ 0.005, two-tailed Fisher's exact test, $N = 40$–47 neurons. **(M)** Length of unpruned dendrites in I–K. Values are mean ± SEM, ***$P < 0.0005$, Wilcoxon-Mann-Whitney test. Scale bars in A, F, and I are 50 μm.

Neurite fragmentation is a hallmark of many pruning processes, and c4da neuron dendrite severing can be seen as an extreme example of fragmentation. The involvement of mechanical forces in other pruning models has not been investigated. For example, pruning of *Drosophila* mushroom body γ neuron axons requires phagocytic glia (Awasaki et al., 2006; Hoopfer et al., 2006), and forces caused by phagocytic engulfment could act to actively break pruning axons. Peripheral sensory neurite morphogenesis in mammals also involves pruning (Meltzer et al., 2022). It is interesting to speculate that movement-induced mechanical forces could play an important role here as well. Thus, the involvement of mechanical force during neurite pruning could be a widespread mechanism.

## Materials and methods
### Fly strains
C4da neurons were labeled by UAS-CD8::GFP expression under *ppk-GAL4* (Grueber et al., 2007) or by *ppk-eGFP* (Grueber et al., 2003b), *ppk-CD4::tdGFP*, *ppk-CD4::tdtomato* (Han et al., 2011), or *ppk-MApHS* (Han et al., 2011) promotor fusions. Other GAL4 lines were *A58-GAL4* (Galko and Krasnow, 2004) and *CCAP-GAL4* (BL25686). For ubiquitous expression with exception of c4da neurons, *act5C-GAL4* (BL4414) was recombined with ppk-GAL80 (Yang et al., 2009). C1da neurons were labeled with *2-21-GAL4* (Grueber et al., 2003a) or through *lexAop-CD8::GFP* (BL32203) expression under *nsyb-lexA* (BL52247). Epithelial cell boundaries were labeled with ECad::GFP (BL60584). The following dsRNA lines were used: Orco (BL 31278) as control, Sox14 (BL 26221), and E74 (VDRC 105301). All dsRNAs were coexpressed with UAS-dcr2 (Dietzl et al., 2007). Other UAS transgenes were UAS-mito::GFP (BL 8442), UAS-hid, UAS-GFP, UAS-palmKate (BL 86540). Mutant alleles were E74$^{DL-1}$ (BL 4435), E74$^{v4}$ (BL 5050), *drpr$^{\Delta 5}$* (Freeman et al., 2003), *wsp$^1$* (BL 51657), and *swp2$^{MICAL}$* (Terman et al., 2002).

### Microscopy and time lapse imaging
C4da neurons (ddaC—dorsal, v'ada—ventrolateral, vdaB—ventral) and the dorsal c1da neurons ddaD and ddaE were imaged in segments A2–A5. To analyze movements of c4da neuron cell bodies and distinct tissue movements between 0 and 12 h APF, animals were imaged on an inverted Zeiss Axio Observer Z1 microscope with a Yokogawa CSU X-1 spinning disk unit in a Zeiss Tempcontrol 37-2 temperature control chamber set to 25°C in 5-min intervals with a 25× Zeiss Plan Apochromat oil objective (0.8 NA) and Zeiss Black software. Focus was adjusted every 1–2 h. Severing events were recorded on an inverted Leica SP8 confocal microscope with a HC PL APO 40×/1.3 NA oil objective in 1-min intervals. For analysis of fast movements, c4da neurons

and epidermal cells expressing mito-GFP were imaged in a single plane at 1 Hz on an inverted Leica SP8 microscope with a 40× objective using LAS X software. For timecourse analysis of dendrite pruning, appropriately staged animals with fluorescently labeled da neurons were imaged live on a Zeiss LSM710 microscope with a 20× Plan Apochromat water objective (1.0 NA) and Zeiss Black software. For analysis of dendrite pruning phenotypes after 14 h APF, animals were dissected out of the pupal case and imaged live on a Zeiss LSM710 with a 20× Plan Apochromat water objective. Immunofluorescence images were obtained on a Zeiss LSM 710 microscope using a 40× Zeiss C Apochromat water objective (1.1 NA). Microscopic images shown are maximum projections or (where indicated) single plane images, all processing was done in Fiji (Schindelin et al., 2012).

### Movement analyses and tracing
C4da neuron cell bodies in long-term time lapse videos were traced in Fiji (Schindelin et al., 2012) using the TrackMate (v3.5.1) plugin. Tracks were checked manually for connection of all soma positions and the displacement values (distance to previous tracking point) were analyzed. Dendrite severing events in time lapse analyses were identified manually.

For analyses of differential tissue movements, c4da neuron cell bodies, dendrite branchpoints, and epidermal cell tricellular junctions were traced manually using the Fiji plugin MTrackJ. Soma-epidermis and dendrite-epidermis distances were normalized to the distance at $t_0$. Data obtained from several experiments were compared by generating Gardner-Altman estimation plots (Ho et al., 2019).

For speckle analysis of fast movements, corresponding vector fields were quantified using spatiotemporal crosscorrelation spectroscopy (Ashdown et al., 2015), using the following parameter settings: pixelSize = 0.45; timeFrame = 1; tauLimit = 9; filtering = FourierWhole; MoveAverage = 5; fitRadius = 8; omegaThreshold = 2; thresholdVector = 8; ROIsize = 16, ROIshift = 4, TOIsize = 5, and TOIshift = 2. Using these parameters, each image of the movie is divided into quadrants of 16 × 16 pixels. For each quadrant, a separate vector is calculated by measuring the object displacement between the initial frame ($n$) and the same quadrant five frames later ($n + 5$). Analysis of every second image (i.e., [1 vs. 6], [3 vs. 8], [5 vs. 10], etc.) is displayed in the videos.

### Mechanical stress setup
To assess viscoelastic properties of c4da neuron dendrites, early pupae (0 h or 5 h APF) were glued with their ventral side to a coverslip which was put on a slide, and dorsal c4da neurons were imaged on a Zeiss LSM 710 microscope. The coverslip was then placed in an ice-cooled water-filled beaker using a custom-

made carrier device and sonicated at 1 cm distance for 30 s with a UP 100H ultrasound processor (dr. hielscher GmbH; pulse amplitude: 100, cycle: 0,6). After the treatment, the coverslip was removed from the beaker, and neurons were imaged again as before. To prevent degeneration, the procedure was carried rapidly within ~15 min. Animals with obvious tissue damage were excluded, and assessed breaks were verified by tracing within Z stacks.

### Immunofluorescence
Antibodies against Mical (1:500, Rode et al., 2018), Sox14 (1:30, Ritter and Beckstead, 2010) and futsch/22C10 (1:50, DGRC) were used as described (Rumpf et al., 2014; Herzmann et al., 2017). Briefly, appropriately staged pupal filets were fixed in 4% formaldehyde, blocked in PBS with 0.3% Triton X-100 and 10% goat serum, and incubated with antibody solutions in blocking buffer overnight. C4da neurons labeled by *ppk* promotor fusions were counterstained with chicken anti-GFP (1:500; Aves labs) or rabbit anti-DsRed antibodies (1:1,000; Clontech).

### Western blot
Wild type FLAG-tagged Par-1 (isoform RR) or the phosphomutant Par-1$^{T636A}$ were expressed in S2R+ cells by cotransfection of the corresponding pUAST expression plasmids (Herzmann et al., 2017) with Actin5C-GAL4. After 72 h, 20 μM 20-hydroxy-ecdysone was added to the medium for 3 h. Cells were harvested in ice-cold PBS and lysed in SDS sample buffer. Lysates were run on 8% gels and blotted with antibodies against phosphorylated Par-1 (phospho-MARK family, #4836; 1:1,000; Cell Signaling Technology). FLAG M2 (F3165; 1:5,000; Sigma-Aldrich) and *Drosophila* VCP (Rumpf lab, raised against recombinant VCP in rabbits at Pineda antibody service, 1:5,000) as a loading control on an Amersham Imager 680.

### Quantification and statistical analyses
Phenotypic penetrance was assessed by counting the number of neurons with dendrites still attached to the soma. Here, significance was determined using a categorical two-tailed Fisher's exact test (graphpad.com). Length of unpruned dendrites were measured using the Fiji NeuronJ plugin and compared using the Wilcoxon Mann Whitney test (Marx et al., 2016).

### Online supplemental material
Fig. S1 shows a Western blot analysis of Par-1 levels and active site phosphorylation induced by ecdysone. Fig. S2 includes stills of individual severing events from confocal time lapse Videos 1 and 2 (1 frame/min). Fig. S3 shows examples of soma—dendrite branchpoint displacement along AP and DV axes. Fig. S4 shows head eversion defects upon E74 and CCAP neuron manipulations and verification of the act-GAL4, ppk-GAL80 line. Fig. S5 shows stainings of the microtubule-associated protein futsch in lateral c4da (v'ada) neurons at the early pupal stage in animals with ablated CCAP neurons. Videos 1 and 2 are confocal time lapse movies of individual severing events (1 frame/min). Video 3 shows a spinning disc time lapse analysis with c4da neuron soma tracing to visualize tissue movements (1 frame/5 min). Video 4 shows a spinning disc time lapse analysis

with tracings of c4da neuron soma and dendrite to visualize differential movements (1 frame/5 min). Videos 5, 6, and 7 show long term spinning disc time lapse analyses with c4da neuron soma tracings to visualize tissue movements during pupal ecdysis in controls and animals lacking E74 or with ablated CCAP neurons (1 frame/5 min). Video 8 shows a time-lapse analysis (10 min, 1 frame/s) of a c4da neuron (labeled by *ppk-CD4::tdtomato*) and epidermal cells labeled by *UAS-mito::GFP* that was used for cross-correlation analyses in Fig. 4.

## Acknowledgments
We thank C. Klämbt for support and comments on the manuscript, K. Franze for helpful discussions, M. Galko, S. Luschnig, P. Soba, C. Wegener, T. Reiff, B. White, C. Han, H. Aberle, the Bloomington, DGRC and VDRC stock centers for fly lines and reagents, and U. Gigengack for expert technical help. R. Krämer and N. Wolterhoff are members of the CRC1348 graduate school.

This work was supported by the DFG Excellence Cluster "Cells in Motion" (EXC1003) (pilot project PP-2017-03) to R. Krämer and M. Galic, the Collaborative Research Center CRC1348 (grant B04 to S. Rumpf), and DFG grant RU1673/3-1 to S. Rumpf.

Author contributions: R. Krämer and N. Wolterhoff performed all time-lapse and mechanical stability experiments and most phenotypic analyses. S. Rumpf contributed to phenotypic analyses. R. Krämer, N. Wolterhoff, and S. Rumpf designed contributed fly lines. M. Galic analyzed movement experiments. All authors contributed to experimental design. S. Rumpf wrote the manuscript with input from all other authors.

Disclosures: The authors declare no competing financial interests exist.

Submitted: 3 May 2022

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

# Supplemental material

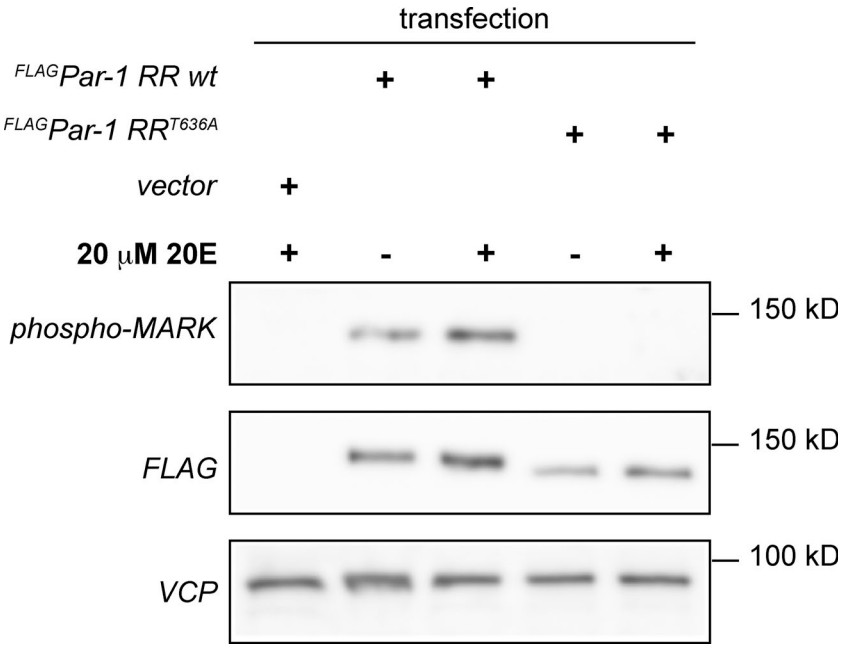

Figure S1.  **Related to** Fig. 1. Increased Par-1 levels and active site phosphorylation in response to ecdysone. S2R+ cells transfected with an empty control plasmid or expression plasmids for FLAG-tagged wild type Par-1 or the non-phosphorylatable mutant Par-1 T636A were treated with 20-hydroxy-ecdysone (20E) for 3 h, lysed and analyzed by Western blot with antibodies against the phosphorylated Par-1 activation loop (phospho-MARK), FLAG or VCP as a loading control (lower blot). Running positions of molecular weight markers are indicated on the right. Source data are available for this figure: SourceData FS1.

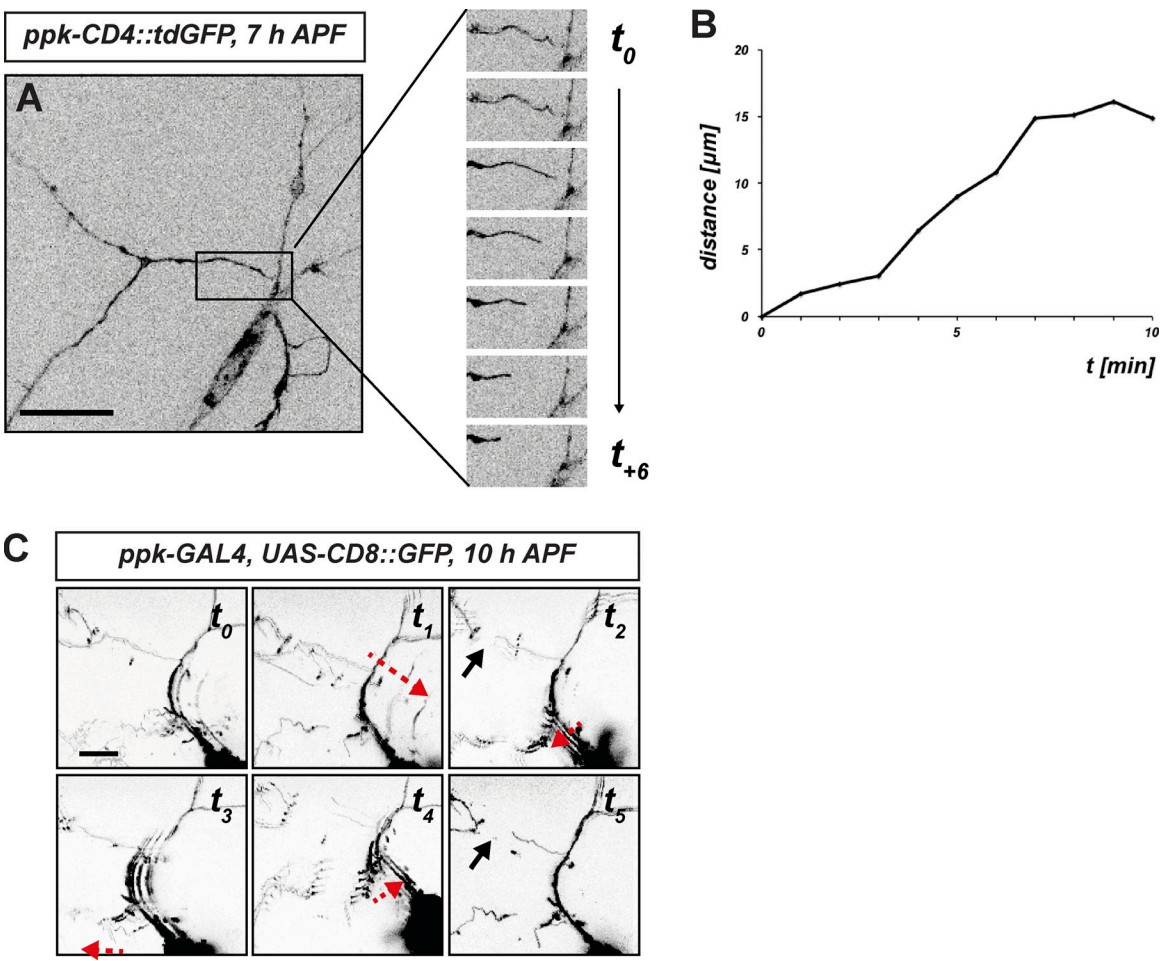

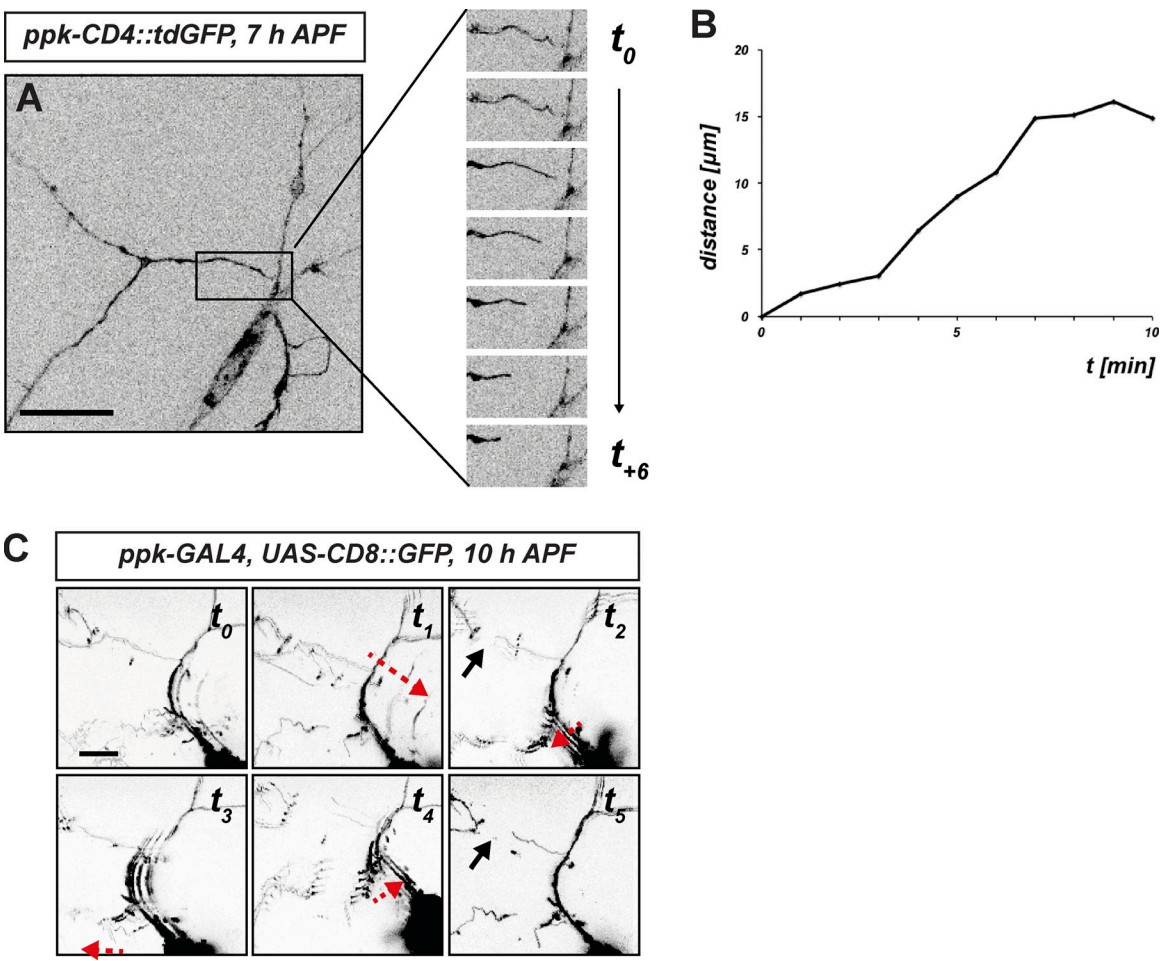

**Figure S2.   Related to** Fig. 2**.** Examples for c4da neuron dendrite severing events with evidence of mechanical tearing. **(A)** Still images of a live movie showing a dendrite severing event. The time series to the right shows the boxed area in the left image in 1-min intervals. **(B)** Distance between the severed dendrite ends in A over time. **(C)** Still images of a similar time series covering a dendrite severing event with evidence of fast tissue movements. Black arrows indicate the position of a recently severed dendrite, red arrows indicate the direction of tissue movement. Scale bars are 50 µm in A and 10 µm in C.

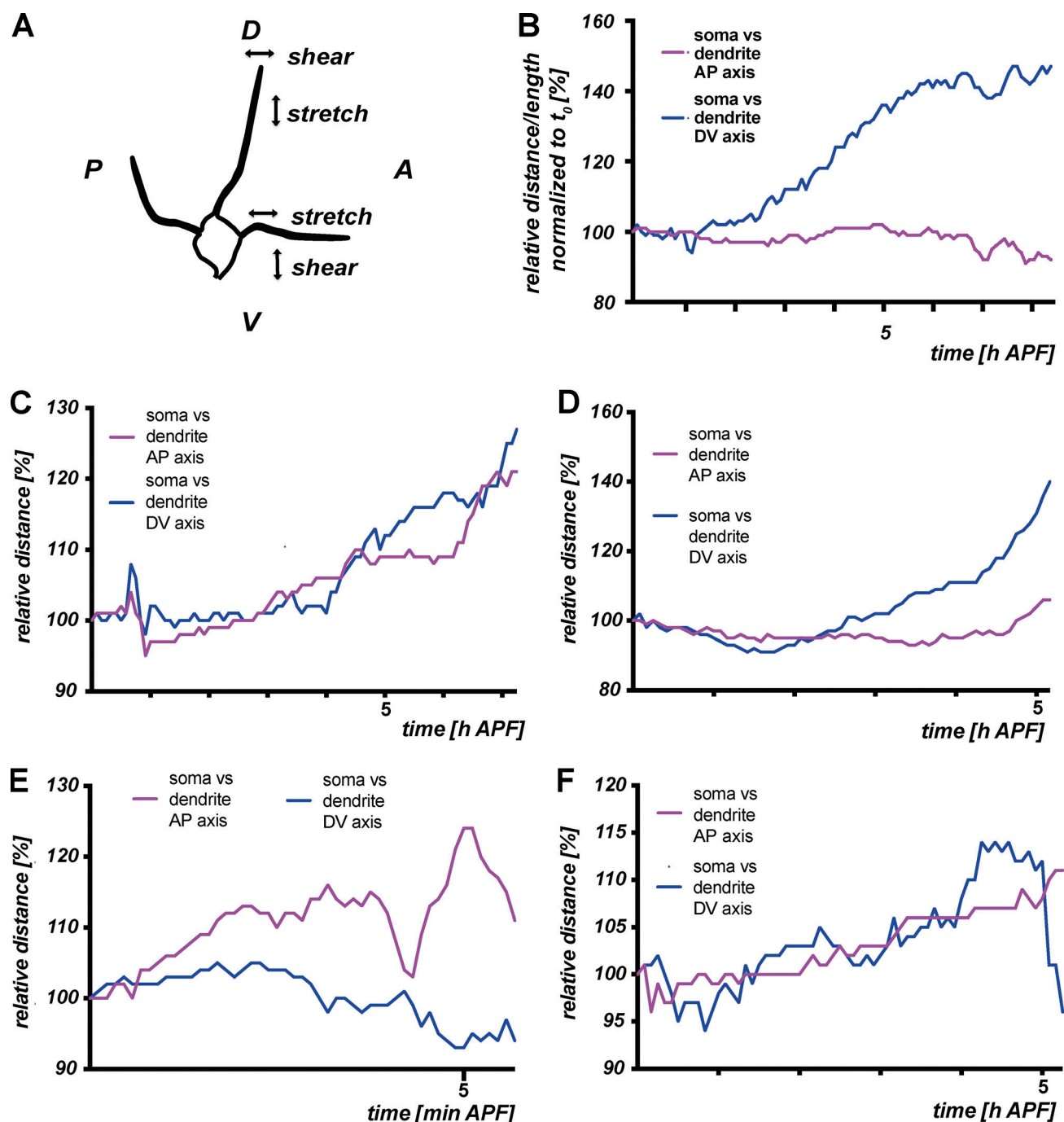

Figure S3. **Examples of c4da neuron cell body movements along the anterior-posterior and dorsoventral axes during the early pupal stage.** **(A)** Schematic representation of how differential soma movement along the anterior-posterior (AP) and dorsoventral (DV) body axes affects dendrites. **(B)** Graph shows AP and DV movement patterns from the time-lapse analysis in Fig. 2, C and D until the time of severing of the marked dendrite. **(C–F)** Graphs show AP and DV movement patterns from other time-lapse analyses performed as in Fig. 2 C (until the time of severing).

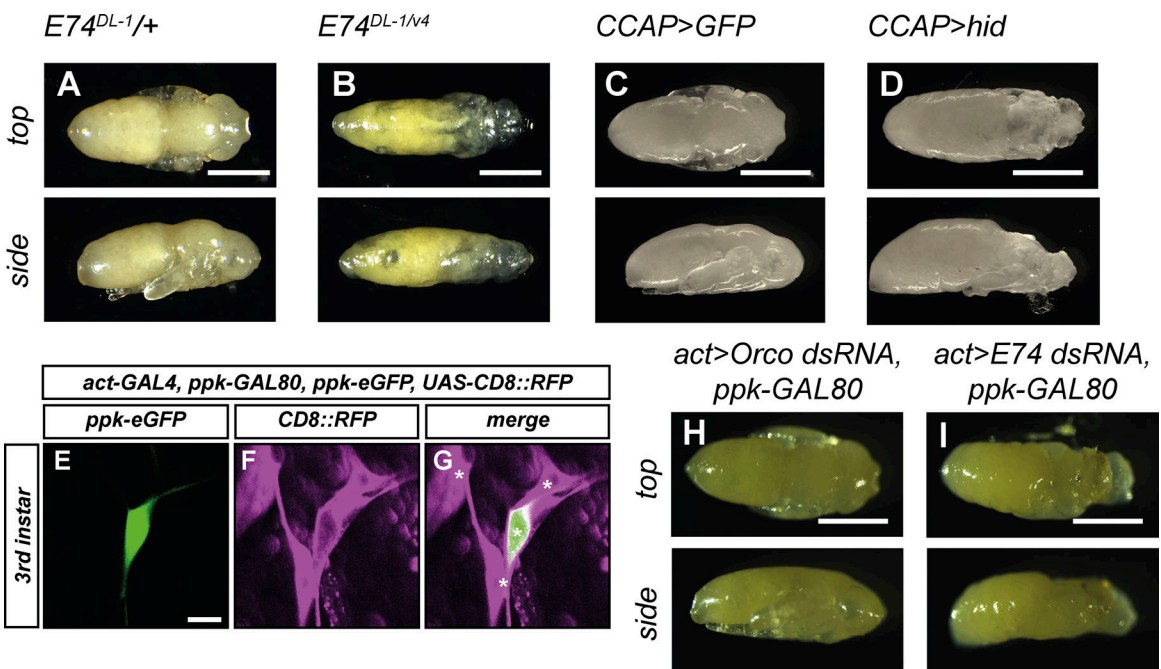

Figure S4. **Ecdysis/head eversion defects upon loss of E74 and upon CCAP neuron ablation.** Animals of the indicated genotypes were dissected out of the pupal case at 18 h APF and photographed from the top (upper panels) and side (lower panels). Note smaller heads, lack of legs, and longer abdomens in experimental genotypes, indicative of ecdysis defects. **(A)** Control animal heterozygous for E74$^{DL-1}$. **(B)** E74$^{DL-1/v4}$ mutant animal. **(C)** Control animal expressing GFP in CCAP neurons. **(D)** Animal with ablated CCAP neurons (hid expression under CCAP-GAL4). **(E–G)** The GAL4 line act-GAL4, ppk-GAL80/CyOweeP; ppk-eGFP (III) was crossed to UAS-mCD8::RFP and dorsal cluster sensory neurons were imaged at the third instar larval stage. A single confocal slice is shown for clarity. **(E)** C4da neuron labeled by ppk-eGFP. **(F)** Dorsal cluster da neurons labeled byCD8::RFP expression under act-GAL4. **(G)** Merge. Positions of da neuron cell bodies are indicated by asterisks. **(H and I)** Pupae ubiquitously expressing the indicated dsRNAs with exception of c4da neurons (act-GAL4, ppk-GAL80). **(H)** Control animal ubiquitously expressing Orco dsRNA. **(I)** Animal expressing E74 dsRNA. Scale bars are 1 mm in A–D, H, and I and 10 μm in E.

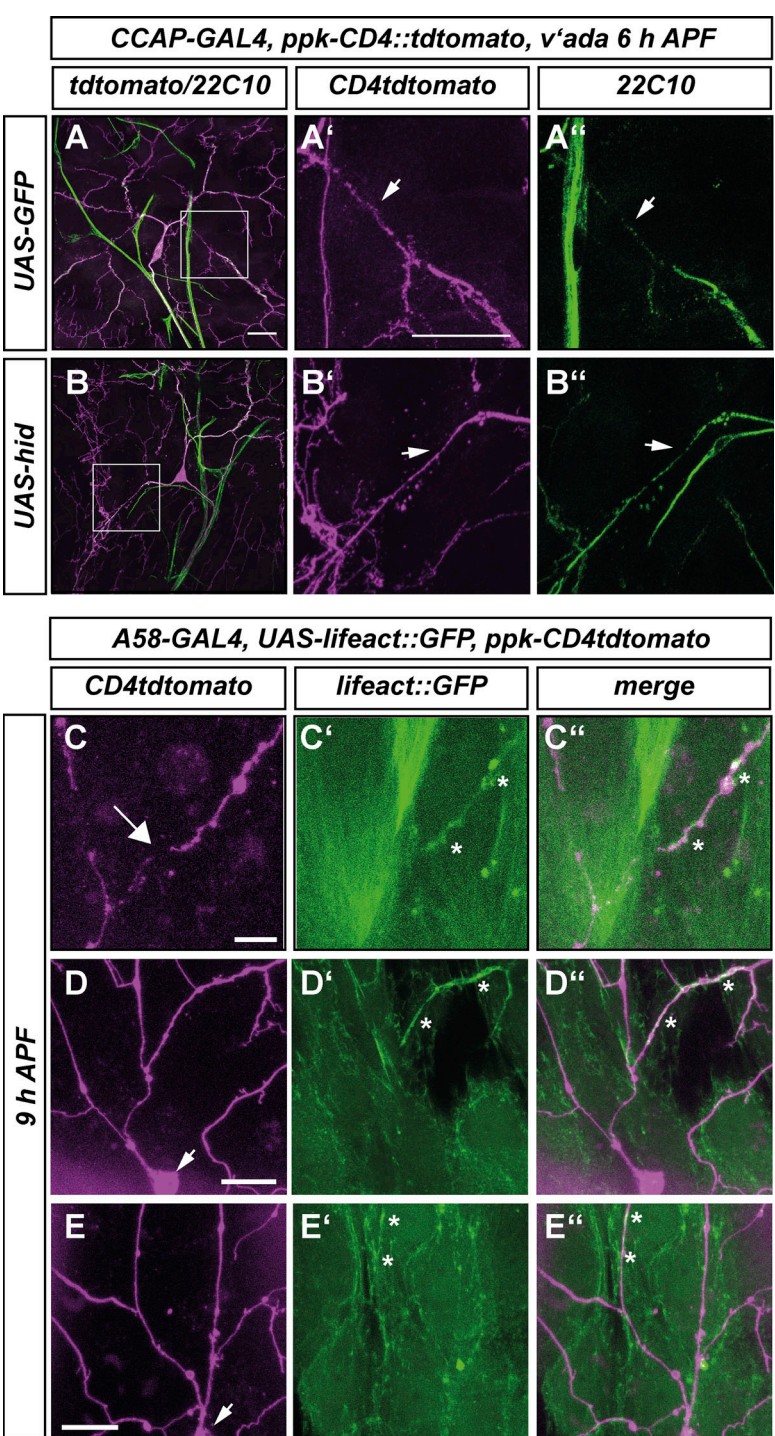

Figure S5. **Cytoskeletal structures in pupal dendrites and epidermis. (A and B)** Effect of CCAP neuron ablation on dendritic microtubule disassembly at 6 h APF in lateral c4da neurons (v'ada). v'ada neurons were visualized with ppk-CD4::tdtomato, and microtubules were stained with anti-futsch (22C10) antibodies. A' and B' show v'ada dendrite regions marked in A and B; A" and B" show the corresponding futsch/22C10 stainings. **(A–A")** v'ada in control animal expressing GFP in CCAP neurons. **(B–B")** v'ada in animal with ablated CCAP neurons. **(C–E)** Epidermal actin structures surround both severed and non-severed c4da neuron dendrites. C4da neurons were visualized with ppk-CD4::tdtomato at 9 h APF. **(C'–E')** show the F-actin reporter UAS-lifeact::GFP under control of the epidermal driver A58-GAL4. **(C"–E")** Merge. **(C–C")** Example of epidermal actin encasing a severed c4da neuron dendrite. **(D–D" and E–E")** Examples of epidermal actin encasing c4da neuron dendrites still attached to the cell body. The arrow in C indicates a severing site, arrows in D and E indicate the position of the cell body. Asterisks in C'–E' and C"–E" indicate close apposition between dendrites and epidermal actin. Scale bars are 10 µm in A, A', and C and 20 µm in D and E.

Video 1.   **Example of snap-back movement of a severed dendrite.** A c4da neuron labeled with *ppk-CD4::tdGFP* was imaged for 30 min in 1-min intervals starting at 7 h APF. A dendrite severing event at ~20 min is followed by snap-back of the severed dendrite. See also Fig. S2 for quantification.

Video 2.   **Example of a c4d neuron dendrite severing event with concomitant tissue movement.** A c4da neuron labeled with *UAS-CD8::GFP* under the control of *ppk-GAL4* was imaged for 10 min in 1-min intervals at 10 h APF. See also Fig. S2 for still images.

Video 3.   **Tracing of c4da neuron cell body during the early pupal stage to illustrate tissue movement.** A c4da neuron labeled with *ppk-eGFP* was imaged in 5-min intervals between 0 and 6.5 h APF. Cell body position over time is indicated by a color trace. Epidermal cells are labeled with *ECad::GFP*. See also Fig. 2, A–A‴ and B for still images and quantification.

Video 4.   **Tracing of c4da neuron cell body, dendrite landmark, and epidermal cell during the early pupal stage to quantify differential movements.** A c4da neuron labeled with *ppk-eGFP* was imaged in 5-min intervals between 0 and 8.3 h APF. Positions of the cell body (red), dendrite branchpoint (blue), and epidermal cell tricellular junction (yellow) over time are indicated by color traces. Epidermal cells are labeled with *ECad::GFP*. See also Fig. 2, C–G for still images and quantification.

Video 5.   **Tracing of control c4da neuron cell body between the onset of the pupal stage until ecdysis.** A c4da neuron labeled with *UAS-CD8::GFP* under the control of *ppk-GAL4* in a *E74^DL-1^/+* control background was imaged in 5-min intervals between 0 and 11 h APF. Positions of the cell body over time are indicated by a red trace. See also Fig. 4, A and D for still image and quantification.

Video 6.   **Tracing of c4da neuron cell body in the *E74* ecdysis mutant between the onset of the pupal stage until ecdysis.** A c4da neuron labeled with *UAS-CD8::GFP* under the control of *ppk-GAL4* in a *E74^DL-1/v4^* mutant background was imaged in 5-min intervals between 0 and 11 h APF. Positions of the cell body over time are indicated by a red trace. See also Fig. 4, B and D for still image and quantification.

Video 7.   **Tracing of c4da neuron cell body in animal with ablated CCAP neurons between the onset of the pupal stage until ecdysis.** A c4da neuron labeled with *ppk-CD4::tdtomato* in an animal expressing *UAS-hid* under *CCAP-GAL4* was imaged in 5-min intervals between 0 and 10 h APF. Positions of the cell body over time are indicated by a red trace. See also Fig. 4, C and D for still image and quantification.

Video 8.   **Fast single-plane video analysis of c4da neuron and epidermal cell movement during ecdysis.** A c4da neuron labeled with *ppk-CD4::tdtomato* in an animal expressing *UAS-mito::GFP* under the epidermal driver *A58-GAL4* was imaged in 1-s intervals for 10 min at 10 h APF. Epidermal mitochondria patterns were then used for spatiotemporal image correlation. See also Fig. 4, F and F′.

