## [Peer Review File · The Journal of Cell Biology]

Developmental Pruning of Sensory Neurites by Mechanical Tearing in *Drosophila*

Rafael Krämer, Neele Wolterhoff, Milos Galic, and Sebastian Rumpf

Corresponding Author(s): Sebastian Rumpf, University of Münster

Review Timeline:

Submission Date:	2022-05-03
Editorial Decision:	2022-05-08
Revision Received:	2022-08-24
Editorial Decision:	2022-10-12
Revision Received:	2022-12-18

Monitoring Editor: Marc Freeman

Scientific Editor: Tim Fessenden

Transaction Report:

DOI: <https://doi.org/10.1083/jcb.202205004>

May 8, 2022

Re: JCB manuscript #202205004

Dr. Sebastian Rumpf
University of Münster
Institute for Neurobiology
Badestrasse 9
Münster 48149
Germany

Dear Dr. Rumpf,

Thank you for submitting your Article manuscript entitled "Developmental Pruning of Sensory Neurites by Mechanical Tearing in *Drosophila*" to Journal of Cell Biology. As part of our normal reviewing procedure, your paper has been evaluated by at least two editors and an editorial statement is provided below. You will see that, in the consensus opinion of our editors, the manuscript is not a good fit for Journal of Cell Biology. We have thus decided not to subject your manuscript to a lengthy external review process. Because Journal of Cell Biology addresses a wide and diverse audience of cell biologists, we must give priority to manuscripts that provide a substantial advance of broad appeal to the cell biology community, even though many others also present interesting and important advances for researchers in a particular field.

We feel your manuscript may be appropriate for Molecular Biology of the Cell or Journal of Cell Science. Although we have not discussed your paper with editors at these journals, you will find the option to easily transfer your manuscript files to either journal at the link.

Link Not Available

I am sorry that our answer on this occasion is not more positive, and I hope that this outcome will not dissuade you from submitting other manuscripts to us in the future.

Thank you for your interest in Journal of Cell Biology.

With kind regards,

Jodi Nunnari
Editor-in-Chief
Journal of Cell Biology

Editorial Statement:

In this work, the authors provide data which suggest that the body movements in moulting *Drosophila* collaborate with sensory neurons to sever dendrites, participating the the pruning process of neural development. Data provided suggest that weakened dendrites bound to epithelial cells are stretched or sheared to sever them from the neuron body, and subsequently are engulfed by macrophages. This work proposes an intriguing integration of physical deformations spanning the organism with cell biological processes to promote proper neural development, and these findings will garner interest among specialists in this field. However, this work is missing mechanistic details to link previously published weakening of dendrite branch points via actin and microtubules with the signaling for body movements. In addition, the presence of stretching and shearing deformations are shown without further details on how these differentially contribute to dendrite detachment. Nor is it clear exactly how macrophage activity promotes shearing, vs simply engulfing cell debris which is an expected for this cell type. Owing to these issues, this work does not convey a significant advance that would interest a broad cell biology readership. If the authors are able, in due time, to resolve these issues in a thoroughly revised manuscript that connects body movements explicitly with established mechanisms of dendrite branch weakening, we would be happy to reconsider this work.

October 12, 2022

RE: JCB Manuscript #202205004R-A

Dr. Sebastian Rumpf
University of Münster
Institute for Neurobiology
Badestrasse 9
Münster 48149
Germany

Dear Dr. Rumpf:

Thank you for submitting your revised manuscript entitled "Developmental Pruning of Sensory Neurites by Mechanical Tearing in *Drosophila*". We would be happy to publish your paper in JCB pending final revisions necessary as requested by reviewers, as well as formatting to meet our guidelines (see details below).

As you will see, all reviewers agree the work is conceptually exciting, based on its generally convincing proposition that mechanical tearing is important for dendrite pruning. Reviewers made suggestions to improve the clarity of some methods, and to better align the conclusions reached with the data shown. Reviewer 3 made a suggestion for an additional experiment to clearly demonstrate causality of muscle movements in pruning. However, this and other experimental data requested can be addressed with changes to the text.

A. MANUSCRIPT ORGANIZATION AND FORMATTING:

- 1) Text limits: Character count for Articles is < 40,000, not including spaces. Count includes abstract, introduction, results, discussion, and acknowledgments. Count does not include title page, figure legends, materials and methods, references, tables, or supplemental legends.
- 2) Figures limits: Articles may have up to 10 main text figures.
- 3) Figure formatting: Scale bars must be present on all microscopy images, including inset magnifications. Molecular weight or nucleic acid size markers must be included on all gel electrophoresis.
- 4) Statistical analysis: Error bars on graphic representations of numerical data must be clearly described in the figure legend. The number of independent data points (n) represented in a graph must be indicated in the legend. Statistical methods should be explained in full in the materials and methods. For figures presenting pooled data the statistical measure should be defined in the figure legends. Please also be sure to indicate the statistical tests used in each of your experiments (either in the figure legend itself or in a separate methods section) as well as the parameters of the test (for example, if you ran a t-test, please indicate if it was one- or two-sided, etc.). Also, if you used parametric tests, please indicate if the data distribution was tested for normality (and if so, how). If not, you must state something to the effect that "Data distribution was assumed to be normal but this was not formally tested."
- 5) Abstract and title: The abstract should be no longer than 160 words and should communicate the significance of the paper for a general audience. The title should be less than 100 characters including spaces. Make the title concise but accessible to a general readership.
- 6) Materials and methods: Should be comprehensive and not simply reference a previous publication for details on how an experiment was performed. Please provide full descriptions in the text for readers who may not have access to referenced manuscripts.
- 7) Please be sure to provide the sequences for all of your primers/oligos and RNAi constructs in the materials and methods. You must also indicate in the methods the source, species, and catalog numbers (where appropriate) for all of your antibodies. Please also indicate the acquisition and quantification methods for immunoblotting/western blots.

8) Microscope image acquisition: The following information must be provided about the acquisition and processing of images:

- Make and model of microscope
- Type, magnification, and numerical aperture of the objective lenses
- Temperature
- Imaging medium
- Fluorochromes
- Camera make and model
- Acquisition software
- Any software used for image processing subsequent to data acquisition. Please include details and types of operations involved (e.g., type of deconvolution, 3D reconstitutions, surface or volume rendering, gamma adjustments, etc.).

10) Supplemental materials: There are strict limits on the allowable amount of supplemental data. Articles may have up to 5 supplemental figures. Please also note that tables, like figures, should be provided as individual, editable files. A summary of all supplemental material should appear at the end of the Materials and methods section.

13) ORCID IDs: ORCID IDs are unique identifiers allowing researchers to create a record of their various scholarly contributions in a single place. At resubmission of your final files, please consider providing an ORCID ID for as many contributing authors as possible.

Please note that JCB now requires authors to submit Source Data used to generate figures containing gels and Western blots with all revised manuscripts. This Source Data consists of fully uncropped and unprocessed images for each gel/blot displayed in the main and supplemental figures. Since your paper includes cropped gel and/or blot images, please be sure to provide one Source Data file for each figure that contains gels and/or blots along with your revised manuscript files. File names for Source Data figures should be alphanumeric without any spaces or special characters (i.e., SourceDataF#, where F# refers to the associated main figure number or SourceDataFS# for those associated with Supplementary figures). The lanes of the gels/blots should be labeled as they are in the associated figure, the place where cropping was applied should be marked (with a box), and molecular weight/size standards should be labeled wherever possible.

WHEN APPROPRIATE: The source code for all custom computational methods published in JCB must be made freely available as supplemental material hosted at www.jcb.org. Please contact the JCB Editorial Office to find out how to submit your custom macros, code for custom algorithms, etc. Generally, these are provided as raw code in a .txt file or as other file types in a .zip file. Please also include a one-sentence summary of each file in the Online Supplemental Material paragraph of your manuscript.

B. FINAL FILES:

Thank you for this interesting contribution, we look forward to publishing your paper in Journal of Cell Biology.

Sincerely,

Marc Freeman
Monitoring Editor
Journal of Cell Biology

Tim Fessenden
Scientific Editor
Journal of Cell Biology

Reviewer #1 (Comments to the Authors (Required)):

In this manuscript, Kramer and colleagues focused on the mechanical stress driving the severing of ddaC dendrites during metamorphosis. Supported by multiple live-imaging and tracing methods, the authors tried to prove the existence of mechanical force between the soma and dendrites at early pupal stage. They also showed that the pupal body movement increases during ecdysis, which may be the cause of mechanical force driving dendrite pruning. And delaying the ecdysis process also postpones the pruning process. Finally, their genetic interaction data suggests the synergy between extrinsic and internal pruning pathways. Overall, their findings are interesting and important, and the story is well written in a logical way. The images and data support most of their key conclusions although they need to tune down some points. I think that the manuscript should be published in JCB as soon as possible. There are some important questions and gaps unsolved in the current study. The authors can consider the following concerns and comments in their future studies.

Concerns and comments:

1. In Fig 1, *sox14* dsRNA was used to show that ecdysone signalling is required to destabilise dendrites mechanically. Although it would not affect the conclusion in any way, it might be interesting to see if the extent of dendrite thinning would be similar between *sox14* dsRNAi and the control. The reduction in the number of breaks/apparent increase in dendrite stability might just be a consequence of a lack of dendrite thinning.
2. In Fig 2, using the tricellular junction appropriate as an anchor control in this experiment. If we look closely at the Ecad-GFP movie (Supplementary Movie 4), the epidermal cells also migrate towards the ventral side, in line with ddaC soma. This raises a question whether the difference of their speed causes the mechanical stress. The authors may discuss a bit.
3. Fig 4E-F show that "ecdysis-induced movement generates local stretch and shear in the vicinity of c4da neurons". These images are very nice and informative. It would be nice to include E74 mutant and CCAP>hid data to show if ecdysis-dependent

movement is required to generate the local forces at the epidermis. But these are not essential for the publication. They can include if the authors have the results. Those would be strong evidence.

4. The current study only suggests a mechanical mechanism during dendrite pruning. To provide direct evidence, the authors can consider recapitulating the ecdysis tissue movement at larval stages (perhaps with magnetic beads) to show that the force acting on the neurons is sufficient to induce dendrite severing. There are still some technical innovations currently. Just for the authors' consideration.

5. One cannot exclude that impaired ecdysis might affect the whole animals at many aspects. It is possible that neuronal functions or ability of epidermal phagocytosis is affected upon loss of E74 or CCAP>hid. The authors should discuss this point, and also tune down their conclusion throughout the paper. For example, "our data establish mechanical tearing as a novel mechanism during neurite pruning." changed to "our data suggest mechanical tearing as a novel mechanism during neurite pruning". There are still some potential gaps in the current study.

Minor points:

1. The measurements of dendrite severing/fragmentation are problematic. Since the authors focused on mechanical force driving dendrite severing, they should look at the severing time point (6-8 h APF) but rather than 16-18 h APF, especially for Fig 5.

2. In Fig S5, the authors just showed their data without any explanation. What is the role F-actin playing at this time point? The authors may provide some speculation in the discussion part. Anyway, this finding is interesting. Please indicate whether the F-actin enrichments in epidermal cells occurred in all neurons (percentage?). Can you see the similar F-actin enrichment in larvae? These questions need to be addressed.

3. "Microtubule dynamics in dendrites are upregulated by the kinase Par-1 during the early pupal stage (Herzmann et al., 2017)". A recent paper on Par-1 overexpression should be cited (Bu S, Cell Rep. 2022). In this study, overexpression of Par-1 is sufficient to impair microtubule polymerization and orientation in the dendrites of ddaC neurons.

4. Fig 3 legend mentions "Arrowheads in A, H and O indicate axons.". But arrowheads are missing in Fig 3O-T. To make them consistent with Figure 3A-F, these arrowheads can be in blue.

Reviewer #2 (Comments to the Authors (Required)):

The removal of dendrites from *Drosophila* sensory neurons during pupariation has been used by several labs to genetically identify regulators of neurite pruning. However, despite the depth of analysis this process has been subject to, the role of mechanical forces has not been investigated. This manuscript from the Rumpf lab beautifully addresses what has apparently been a major gap in our understanding of developmental pruning of dendritic arborization neurons. They show that some of the cell-autonomous pruning regulators identified in this system cause dendrites to become sensitive to mechanical disruption at a specific stage of pupal development. They go on to show that this mechanical sensitivity is functionally important for normal pruning. First, they correlate the timing of pruning with increased movement of the animal and movements of the epidermis and dendrites. Next, they develop methods to block these movements and show that this impairs pruning by affecting non-neuronal cells. In all, they use a set of logical, well-controlled experiments to build a strong argument that forces outside the neurons themselves contribute to developmental pruning. This story fills in an important gap in our understanding of an important model system in which pruning is studied, and provides context for considering whether mechanical forces may also be part of neuronal remodeling in other contexts. I have only a few suggestions for changes:

1. How are breaks defined and validated in the sonication experiment in Figure 1? Because thinning has already begun, fluorescence is dim before sonication, and so the difference between the Orco RNAi before and after sonication looks quite subtle. Would breaks be clearer if a soluble GFP was included with the membrane-bound one?

2. I had some trouble understanding the branch point angle change in Figure 2. If the dendrites are attached to the epidermal cells, how would the angle between branches change? Are the associated epidermal cells also changing shape during this time period in such a way that could account for branching changes in dendrites? Are the epidermal cells also predicted to be under tension? Later some movements are shown in epidermal cells, but it is unclear how this might relate to the idea of tension-related shape changes.

Minor comments

1. The significance of ORCO RNAi in Figure 1 is not explained in the text. I assume this is a control RNAi, but it would be helpful to mention that.

2. some graphs had lines around some sides, but not others- for example in Fig 3. And labeling of graphs could be more consistent throughout.

3. dendrites are sometimes describes as undigested; I am not sure what this means.

4. when class I neurons are analyzed, why is sometimes ddaE chosen (fig 3) and sometimes ddaD (Fig 6)?

Reviewer #3 (Comments to the Authors (Required)):

Krämer et al. studied how mechanical forces contribute to developmental pruning of neuronal dendrites. They identified morphogenetic pupal movement as a driving force for dendrite breakage shortly after the cytoskeletal destabilization phase. First, they show that dendrites are more sensitive to mechanical forces during the pupal stage as sonication of pupae can induce dendrite breakage after the initial cytoskeletal destabilization phase, which is dependent on ecdysone-induced Sox14 activity. They characterized the relative movement of sensory neuron somata and their dendrites during pruning indicating that stretch and shear forces are generated due to distinct relative displacement of the epidermis/dendrites and the soma. Furthermore, the authors show that blocking pupal ecdysis results in reduced dendrite severing, presumably due to a (partial) block of pupal movement. They identify this as a non-autonomous mechanism that synergistically contributes to dendrite pruning together with local pathways including cytoskeletal integrity and phagocytic activity by surrounding epithelial cells. The overall study is conceptually very exciting and provides highly interesting insight into the impact of developmental tissue movement on neurite pruning. Tissue mechanics/movement and its role in development is so far heavily understudied and only few recent studies highlight a critical role in morphogenesis. Thus, this study provides imminent insight into the role of physiological tissue movement and developmental neurite pruning processes in a relevant *in vivo* system.

The only major limitation of this work is that although the authors show that pupal movement, mechanical shear, and sensory neuron displacement during pupal ecdysis correlate with dendrite breakage, it is nonetheless difficult to claim causality. However, I believe the authors can address this point to strengthen the validity of their conclusions. In this regard, I have three suggestions that could help to establish stronger correlation and provide a more causal link.

1. Regarding the sonication parameters, to what kind of mechanical forces do they correspond? Do the pupae survive this treatment? That would be relevant to estimate the biological relevance of this experiment. Along the same lines, it would be meaningful regarding causal role of mechanical forces in dendrite severing if the authors can use a more physiological stimulus, e.g. optogenetic activation of rhythmic muscle contraction.
2. If ecdysis is blocked, can dendrites still be pruned upon mechanical stimulation e.g. by sonication or artificial (optogenetic) muscle activation? This would strengthen the evidence that tissue movement is necessary and pruning defects are not due to secondary effects of the genetic manipulation, e.g. on cytoskeletal structure or due to pupal lethality of CCAP/E74 manipulation.
3. It is interesting that ablation of CCAP-positive cells results in strong pruning defects in lateral and ventral c4da neurons rather than dorsally. A recent study characterized the morphogenetic movements of pupae with a high level of detail (Elliot et al. eLife 2021, doi: 10.7554/eLife.68656). In this study they showed differential contribution of ETH and CCAP signaling to pupal movement during ecdysis ("ETHRB neuron suppression blocks the Lift, a movement of the posterior compartment, while suppressing CCAP neurons prematurely terminates the first (and only) swing-like movement by blocking its progression into the anterior compartment. "). Can the authors identify if the role of CCAP, or more generally the different muscle contraction modes, coincide with the dendrite severing timing of ddac vs v'ada and vdaB? This would strengthen the notion that specific motor-induced movement programs during ecdysis contribute to dendrite severing of the different c4da neurons.

Additional comments:

1. In Fig. 1, the number of neurons with induced breaks after sonication was quantified in controls, but not in the Sox14-RNAi experiment. The number of breaks is statistically increased, but it is unclear if this reflects only the neurons that are already in the process of severing their dendrites or also additional ones.
2. Fig. 2G: based on the legend it is not clear to me what is shown here. I guess the authors refer to the schematic in Fig. S3 in the legend and not to the graph shown in Fig. 2G, which according to the main text indicates the length changes between the soma and the 2nd dendritic branch point. As tissue movement and distortion is different in distinct body wall regions, to me this is the more important and relevant analysis rather than quantifying the distance between the c4da soma or distal dendritic branch points and an epidermal cell further away as shown in Fig. 2C-F. As the breakage typically occurs proximal to the 1st or 2nd branch point as stated by the authors, I would assume the critical force for dendrite tearing is exerted between the glia-wrapped soma and the proximal branch point(s). Can the authors focus on this analysis and provide a relevant example in the figure?
3. In Fig. 2E/F, the "de-ep minus se-ep" graph is not fully clear to me and there is no reference to it in the legend or text. Does this indicate the difference in variability between the neuron and epidermis vs. the dendrite and epidermis distance? It will be helpful to clearly explain this metric.
4. Figure 3 indicates that most dendrites are severed between 10-18h APF, with some earlier events for ddaC neurons, thus corresponding with pupal ecdysis. However, blocking ecdysis by E74-RNAi or using a mutant allele result in fairly weak dendrite

retention. In my opinion, the retained dendrite length is a less meaningful metric than the number of neurons with non-severed dendrites as also shown later, e.g. Fig.6. This is however missing here and should be amended (if 100% it can just be mentioned in the text).

5. In Fig. S3 the authors show examples of c4d soma displacement along the AP/DV axis as evidence for a contribution to dendrite severing. Can they indicate in the graphs at which time point dendrite severing occurs?

6. In Fig. 6K, please indicate that c1da neurons are quantified here (ddaD is mentioned in the legend, but just looking at the figure it seems to be confusing).

Response to reviewers' comments

Manuscript 202205004R-A

Developmental Pruning of Sensory Neurites by Mechanical Tearing in *Drosophila*

Rafael Krämer^{1,4}, Neele Wolterhoff^{1,3,4}, Milos Galic², Sebastian Rumpf^{1*}

Reviewer #1 (Comments to the Authors (Required)):

In this manuscript, Kramer and colleagues focused on the mechanical stress driving the severing of ddaC dendrites during metamorphosis. Supported by multiple live-imaging and tracing methods, the authors tried to prove the existence of mechanical force between the soma and dendrites at early pupal stage. They also showed that the pupal body movement increases during ecdysis, which may be the cause of mechanical force driving dendrite pruning. And delaying the ecdysis process also postpones the pruning process. Finally, their genetic interaction data suggests the synergy between extrinsic and internal pruning pathways. Overall, their findings are interesting and important, and the story is well written in a logical way. The images and data support most of their key conclusions although they need to tune down some points. I think that the manuscript should be published in JCB as soon as possible. There are some important questions and gaps unsolved in the current study. The authors can consider the following concerns and comments in their future studies.

Concerns and comments:

1. In Fig 1, sox14 dsRNA was used to show that ecdysone signalling is required to destabilise dendrites mechanically. Although it would not affect the conclusion in any way, it might be interesting to see if the extent of dendrite thinning would be similar between sox14 dsRNAi and the control. The reduction in the number of breaks/apparent increase in dendrite stability might just be a consequence of a lack of dendrite thinning.

Our response: Thank you for this comment. We agree that this is likely the case - we presume that the thinnings (as defined by Kanamori et al., 2015, Nat. Comm.) are in fact the mechanically unstable sites. To make this link clearer, we counted the thinnings in the Orco dsRNA control and Sox14 dsRNA samples (new Fig. 1 E). This analysis shows that significantly fewer c4da neurons expressing Sox14 dsRNA have thinnings and support the reviewer's notion that the thinnings are the fragile sites.

2. In Fig 2, using the tricellular junction appropriate as an anchor control in this experiment. If we look closely at the Ecad-GFP movie (Supplementary Movie 4), the epidermal cells also migrate towards the ventral side, in line with ddaC soma. This raises a question whether the difference of their speed causes the mechanical stress. The authors may discuss a bit.

Our response: Thank you for this comment. We added clarifying sentences in the corresponding paragraph stating that "The whole tissue underwent movements in these experiments..." and that the asynchronous movement could be "...due to slight differences in speed and/or movement direction" (p. 7).

3. Fig 4E-F show that "ecdysis-induced movement generates local stretch and shear in the vicinity of c4da neurons". These images are very nice and informative. It would be nice to include E74 mutant and CCAP>hid data to show if ecdysis-dependent movement is required to generate the local forces at the epidermis. But these are not essential for the publication. They can include if the authors have the results. Those would be strong evidence.

Our response: Thank you for this comment. This experiment is difficult for technical reasons because the unmarked E74 mutations and the A58-GAL4 driver used are both on the third chromosome, which makes the required recombination extremely challenging.

4. The current study only suggests a mechanical mechanism during dendrite pruning. To provide direct evidence, the authors can consider recapitulating the ecdysis tissue movement at larval stages (perhaps with magnetic beads) to show that the force acting on the neurons is sufficient to induce dendrite severing. There are still some technical innovations currently. Just for the authors' consideration.

Our response: Thank you for this comment. We added the following sentence in the discussion: "To test whether mechanical forces are sufficient to tear dendrites, the effects of optogenetic induction of ecdysis movements at various developmental stages on dendrites could be assessed." (p14).

5. One cannot exclude that impaired ecdysis might affect the whole animals at many aspects. It is possible that neuronal functions or ability of epidermal phagocytosis is affected upon loss of E74 or CCAP>hid. The authors should discuss this point, and also tune down their conclusion throughout the paper. For example, "our data establish mechanical tearing as a novel mechanism during neurite pruning." changed to "our data suggest mechanical tearing as a novel mechanism during neurite pruning". There are still some potential gaps in the current study.

Our response: Thank you for this comment. We changed the above sentence accordingly and added an additional cautionary sentence in the Discussion (p. 14): "While we excluded that our ecdysis manipulations (E74 mutation/CCAP neuron ablation) affect the cell autonomous ecdysone-dependent c4da neuron pruning program, we cannot entirely exclude that they may cause defects in other (unknown) non-cell autonomous pruning pathways."

Minor points:

1. The measurements of dendrite severing/fragmentation are problematic. Since the authors focused on mechanical force driving dendrite severing, they should look at the severing time point (6-8 h APF) but rather than 16-18 h APF, especially for Fig 5.

Our response: Thank you for this comment. We used the 18 h APF endpoint assay here to be able to compare our data with other papers in the field where this is the standard analysis. Pupal ecdysis as an important source of the mechanical forces for dendrite severing only occurs between 10 - 12 h APF, and ecdysis manipulations are not expected to have an effect at the 6 - 8 h APF time window.

2. In Fig S5, the authors just showed their data without any explanation. What is the role F-actin playing at this time point? The authors may provide some speculation in the discussion part. Anyway, this finding is interesting. Please indicate whether the F-actin enrichments in epidermal cells occurred in all neurons (percentage?). Can you see the similar F-actin enrichment in larvae? These questions need to be addressed.

Our response: Thank you for this comment. We are actually referring to an experiment in Han et al., Neuron 2014. These authors showed that pruning c4da neuron dendrites are often ensheathed by epidermal cell F-actin. These authors showed that these structures are induced by the phagocytic receptor Draper, and their occurrence correlates with dendrite fragmentation. However, it was unclear whether the ensheathed dendrites have to be severed from the cell body. We now added a clarifying sentence in the corresponding paragraph:

"Phagocytosis and epidermal cell actin polymerization around severed dendrites have been proposed to contribute to dendrite fragmentation (Han et al., 2014; Williams et al., 2006), but it was unclear whether they could only act on severed dendrites. A cursory investigation of F-actin structures in early pupal epidermis cells showed that actin-rich structures occurred around both severed dendrites and dendrites that were still attached to the soma (Fig. S5), opening up the possibility that local F-actin could also play a role during dendrite severing." (p. 13)

3. "Microtubule dynamics in dendrites are upregulated by the kinase Par-1 during the early pupal stage (Herzmann et al., 2017)". A recent paper on Par-1 overexpression should be cited (Bu S, Cell Rep. 2022). In this study, overexpression of Par-1 is sufficient to impair microtubule polymerization and orientation in the dendrites of ddaC neurons.

Our response: Thank you for this comment. We have now clarified this point in the introduction by stating: "Microtubule dynamics in dendrites are increased during the early pupal stage by fine-tuned activation of the kinase Par-1 (Bu et al., 2022, Herzmann et al., 2017)."

4. Fig 3 legend mentions "Arrowheads in A, H and O indicate axons.". But arrowheads are missing in Fig 3O-T. To make them consistent with Figure 3A-F, these arrowheads can be in blue.

Our response: Thank you for pointing this out. We have now added arrowheads to mark all axons.

Reviewer #2 (Comments to the Authors (Required)):

The removal of dendrites from *Drosophila* sensory neurons during pupariation has been used by several labs to genetically identify regulators of neurite pruning. However, despite the depth of analysis this process has been subject to, the role of mechanical forces has not been investigated. This manuscript from the Rumpf lab beautifully addresses what has apparently been a major gap in our understanding of developmental pruning of dendritic arborization neurons. They show that some of the cell-autonomous pruning regulators identified in this system cause dendrites to become sensitive to mechanical disruption at a specific stage of pupal development. They go on to show that this mechanical sensitivity is functionally important for normal pruning. First, they correlate the timing of pruning with increased movement of the animal and movements of the epidermis and dendrites. Next, they develop methods to block these movements and show that this impairs pruning by affecting non-neuronal cells. In all, they use a set of logical, well-controlled experiments to build a strong argument that forces outside the neurons themselves contribute to developmental pruning. This story fills in an important gap in our understanding of an important model system in which pruning is studied, and provides context for considering whether mechanical forces may also be part of neuronal remodeling in other contexts. I have only a few suggestions for changes:

1. How are breaks defined and validated in the sonication experiment in Figure 1? Because thinning has already begun, fluorescence is dim before sonication, and so the difference between the Orco RNAi before and after sonication looks quite subtle. Would breaks be clearer if a soluble GFP was included with the membrane-bound one?

Our response: Thank you for this comment. It is actually known that soluble fluorescent markers are often excluded from thin neurite structures (such as higher-order dendrites in larval neurons and dendritic thinnings during the pruning phase) (see, e. g., Lee & Luo, *Neuron* 1999). Sonication was carried out in a ice-cooled water bath, and Before/after sonication microscopical analyses were performed within 15 minutes to avoid sample degeneration during the experiment. Breaks were defined as clearly visible gaps dendrites that occurred after sonication. To avoid inadvertent omissions of dendrite stretches, thick Z-stacks were chosen, dendrites were traced in the Z-slices. To address this criticism, we now describe the procedure accordingly in the Methods section.

2. I had some trouble understanding the branch point angle change in Figure 2. If the

dendrites are attached to the epidermal cells, how would the angle between branches change? Are the associated epidermal cells also changing shape during this time period in such a way that could account for branching changes in dendrites? Are the epidermal cells also predicted to be under tension? Later some movements are shown in epidermal cells, but it is unclear how this might relate to the idea of tension-related shape changes.

Our response: Thank you for this comment. C4da neuron dendrites are essentially "sandwiched" between the extracellular matrix (ECM) and the epidermal cells. ECM attachment depends on integrins. At sites where the dendrites are not attached to the ECM, they can "sink into" the epidermis layer and even become embedded inside epidermal cells (in what appears as little tunnels). This deep dendrite attachment to the epidermis affects approximately 5 - 10 % of the dendritic arbor, thus allowing for the shape changes accompanying morphogenesis. In response to the comment, we clarify this now in the introduction:

"whereas the distal dendrites are sandwiched between the extracellular matrix and epidermal cells (Han et al., 2012; Kim et al., 2012). At some sites, contact between dendrites and epidermal cells is very close, such that dendrites seem to "sink into" the epidermal layer. It is estimated that 5 - 10% of the c4da dendritic arbor is enclosed in the epidermis in this way (Han et al., 2012; Kim et al., 2012). (p. 4)

Minor comments

1. The significance of ORCO RNAi in Figure 1 is not explained in the text. I assume this is a control RNAi, but it would be helpful to mention that.

Our response: Thank you for this comment. Orco dsRNA was indeed used as a control for a gene that is not expressed in c4da neurons and is not apparently involved in larval/pupal PNS development. We added an explanatory sentence at the first use (p. 5).

2. some graphs had lines around some sides, but not others- for example in Fig 3. And labeling of graphs could be more consistent throughout.

Our response: Thank you for this comment. We agree that graph design is not entirely consistent, in part due to use of different computer programs (i. e., Excel and Prism). In the graphs in Figure 2, the right and left y axes designate different aspects (relative displacement within one sample vs differences between samples). We reviewed all graphs and labeled them more consistently where possible (Fig. 2 B vs 4 D, 2 D - E).

3. dendrites are sometimes describes as undigested; I am not sure what this means.

Our response: Thank you for this comment. This was meant to describe dendrites that have been severed from the soma but not yet fragmented and phagocytosed. We changed the wording in three instances to be more precise (p. 11, legend Fig. 7).

4. when class I neurons are analyzed, why is sometimes ddaE chosen (fig 3) and sometimes ddaD (Fig 6)?

Our response: Thank you for this comment. We have now added the ddaD quantification as well (new Fig. 3 U), which shows a similar timecourse of dendrite severing as ddaE. ddaE neurons also show pruning defects upon CCAP neuron ablation, but their dendrites are often severed. As we had to use a pan-neuronal driver in this experiment which also labels dendrites from other PNS neurons, it is very difficult to unambiguously assign dendrites to their respective neurons. ddaD dendrites were easier to identify in this experiment. For the sake of clarity, we therefore only used ddaD in this analysis.

Reviewer #3 (Comments to the Authors (Required)):

Krämer et al. studied how mechanical forces contribute to developmental pruning of neuronal dendrites. They identified morphogenetic pupal movement as a driving force for dendrite breakage shortly after the cytoskeletal destabilization phase. First, they show that dendrites are more sensitive to mechanical forces during the pupal stage as sonication of pupae can induce dendrite breakage after the initial cytoskeletal destabilization phase, which is dependent on ecdysone-induced Sox14 activity. They characterized the relative movement of sensory neuron somata and their dendrites during pruning indicating that stretch and shear forces are generated due to distinct relative displacement of the epidermis/dendrites and the soma. Furthermore, the authors show that blocking pupal ecdysis results in reduced dendrite severing, presumably due to a (partial) block of pupal movement. They identify this as a non-autonomous mechanism that synergistically contributes to dendrite pruning together with local pathways including cytoskeletal integrity and phagocytic activity by surrounding epithelial cells.

The overall study is conceptually very exciting and provides highly interesting insight into the impact of developmental tissue movement on neurite pruning. Tissue mechanics/movement and its role in development is so far heavily understudied and only few recent studies highlight a critical role in morphogenesis. Thus, this study provides imminent insight into the role of physiological tissue movement and developmental neurite pruning processes in a relevant in vivo system.

The only major limitation of this work is that although the authors show that pupal movement, mechanical shear, and sensory neuron displacement during pupal ecdysis correlate with dendrite breakage, it is nonetheless difficult to claim causality. However, I believe the authors can address this point to strengthen the validity of their conclusions. In this regard, I have three suggestions that could help to establish stronger correlation and provide a more causal link.

1. Regarding the sonication parameters, to what kind of mechanical forces do they correspond? Do the pupae survive this treatment? That would be relevant to estimate the biological relevance of this experiment. Along the same lines, it would be meaningful regarding causal role of mechanical forces in dendrite severing if the authors can use a more physiological stimulus, e.g. optogenetic activation of rhythmic muscle contraction.

Our response: Thank you for this comment. Due to both the local nature of ultrasonic force generation, and the additional complication through the cuticle/pupal case, it is

not possible to measure the forces generated in this experiment. We performed the sonication on ice and adjusted the sonication time to values that did not destroy pupae (e. g., partial bursting, likely through overheating) and did not cause apparent gross tissue damage in microscopic analyses (both of which happened at longer exposure times). Cursory experiments indicated that some animals survived the treatment. We agree that this experiment is rather crude, and we don't claim that it reflects a physiological condition. However, as the presence of the pupal case precludes any quantitative analyses with direct touch, this is at current the only way to probe dendrite mechanical stability during the early pupal stage. We have tried to clarify this in the description of the experiment:

As the presence of the hardening pupal case precludes any direct quantitative methods during the early pupal stage, we applied mechanical stress through brief sonication in a ice-cooled water bath. (p. 5)

2. If ecdysis is blocked, can dendrites still be pruned upon mechanical stimulation e.g. by sonication or artificial (optogenetic) muscle activation? This would strengthen the evidence that tissue movement is necessary and pruning defects are not due to secondary effects of the genetic manipulation, e.g. on cytoskeletal structure or due to pupal lethality of CCAP/E74 manipulation.

Our response: Thank you for this comment. We don't think that these pruning defects are due to secondary effects for several reasons. Firstly, the effect of both E74 and CCAP neuron manipulations on c4da neuron dendrite pruning is only partial (Figs. 5, 6), likely owing to the effects of the cell-autonomous degeneration program (cytoskeleton disassembly, endocytosis, caspases). Secondly, we also assessed the integrity of the cell-autonomous ecdysone-induced program by staining for the pruning factors Sox14 and Mical, which are still expressed. Thirdly, neurons with ablated CCAP neurons sometimes survive to adulthood (Park et al., 2003), thus making fast pupal lethality at the time of dendrite pruning unlikely. To test for dendritic cytoskeleton disruption in animals with ecdysis defects, we also assessed microtubule content at 6 h APF in lateral c4da neurons upon CCAP neuron ablation (with strong pruning defects). These neurons still had disrupted microtubules in proximal dendrites (new Figure S5 A, B).

3. It is interesting that ablation of CCAP-positive cells results in strong pruning defects in lateral and ventral c4da neurons rather than dorsally. A recent study characterized the morphogenetic movements of pupae with a high level of detail (Elliot et al. eLife 2021, doi: 10.7554/eLife.68656). In this study they showed differential contribution of ETH and CCAP signaling to pupal movement during ecdysis ("ETHRB neuron suppression blocks the Lift, a movement of the posterior compartment, while suppressing CCAP neurons prematurely terminates the first (and only) swing-like movement by blocking its progression into the anterior compartment. "). Can the authors identify if the role of CCAP, or more generally the different muscle contraction modes, coincide with the dendrite severing timing of ddac vs v'ada and vdaB? This would strengthen the notion that specific motor-induced movement programs during ecdysis contribute to dendrite severing of the different c4da neurons.

Our response: Thank you for this comment and the reference. Our notion was very speculative and this reference seems to support it which is great! Unfortunately we do not have a way to image specific ecdysis movements and severing events simultaneously and thus cannot correlate them at this time. We are rephrasing the description of the result accordingly: "In contrast to these relatively mild defects, CCAP neuron ablation caused strong severing defects in the ventrolateral (v'ada) and ventral (vdaB) c4da neurons at 18 h APF (Fig. 6 E - H). Interestingly, it was shown recently that CCAP neuron inhibition blocks a specific part of the ecdysis motor programme (Elliot et al., 2021), such that locally, some movements may be more affected by CCAP ablation than others." (p. 12)

Additional comments:

1. In Fig. 1, the number of neurons with induced breaks after sonication was quantified in controls, but not in the Sox14-RNAi experiment. The number of breaks is statistically increased, but it is unclear if this reflects only the neurons that are already in the process of severing their dendrites or also additional ones.

Our response: Thank you for this comment. We are not sure but this might be a misunderstanding? We assessed the number of dendrite breaks for the Orco control and Sox14 dsRNA at both 0 and 5 h APF, and before and after sonication for both samples at 5 h APF. We now also quantified the appearance of dendritic blebs and thinnings in the 5 h APF samples (new Fig. 1 E). This number is decreased in the Sox14 dsRNA sample, indicating that these thinnings are the primary fragile sites.

2. Fig. 2G: based on the legend it is not clear to me what is shown here. I guess the authors refer to the schematic in Fig. S3 in the legend and not to the graph shown in Fig. 2G, which according to the main text indicates the length changes between the soma and the 2nd dendritic branch point. As tissue movement and distortion is different in distinct body wall regions, to me this is the more important and relevant analysis rather than quantifying the distance between the c4da soma or distal dendritic branch points and an epidermal cell further away as shown in Fig. 2C-F. As the breakage typically occurs proximal to the 1st or 2nd branch point as stated by the authors, I would assume the critical force for dendrite tearing is exerted between the glia-wrapped soma and the proximal branch point(s). Can the authors focus on this analysis and provide a relevant example in the figure?

Our response: Thank you for pointing this out. We forgot to update the legend from an earlier version of the manuscript. The graph is indeed meant to show the lengths of defined proximal dendrite segments at the onset of the pupal stage, and shortly before severing in this region. We now removed Figure 2 E (differential movement prior to 5 h APF). Instead, we added images of the proximal dendrite in the timelapse movie in 2C at 0 h APF and 7.5 h APF (shortly before severing). We marked the measured dendrite segment to enable a length comparison (new Figure 2 F).

3. In Fig. 2E/F, the "de-ep minus se-ep" graph is not fully clear to me and there is no reference to it in the legend or text. Does this indicate the difference in variability between the neuron and epidermis vs. the dendrite and epidermis distance? It will be helpful to clearly explain this metric.

Our response: Thank you for pointing this out. This interpretation is correct. We added the following explanatory sentence in the main text: "These tendencies - synchronous movement between dendrite and epidermis and asynchronous movement between cell body and epidermis - were seen consistently in such timelapse analyses, especially after 5 h APF (Fig. 2 E)."

We also updated the legend accordingly: "E Distribution of cell body/epidermis and dendrite/epidermis distances after 5 h APF normalized to the distance at t_0 (N=10 timelapse movies)."

4. Figure 3 indicates that most dendrites are severed between 10-18h APF, with some earlier events for ddaC neurons, thus corresponding with pupal ecdysis. However, blocking ecdysis by E74-RNAi or using a mutant allele result in fairly weak dendrite retention. In my opinion, the retained dendrite length is a less meaningful metric than the number of neurons with non-severed dendrites as also shown later, e.g. Fig.6. This is however missing here and should be amended (if 100% it can just be mentioned in the text).

Our response: Thank you for this comment. We changed these quantifications in Figs. 5 and 7 now and put the percentages of neurons with dendrites severing defects (i. e., neurons with dendrites attached to the soma).

5. In Fig. S3 the authors show examples of c4d soma displacement along the AP/DV axis as evidence for a contribution to dendrite severing. Can they indicate in the graphs at which time point dendrite severing occurs?

Our response: Thank you for this comment. All measurements actually stop at the time of severing. We have now clarified this both in the main text and the Supplementary Figure legend.

6. In Fig. 6K, please indicate that c1da neurons are quantified here (ddaD is mentioned in the legend, but just looking at the figure it seems to be confusing).

Our response: Thank you for this comment. We added a note in the figure accordingly.